# Human monoclonal antibodies against *Staphylococcus aureus* surface antigens recognize in vitro and in vivo biofilm

**Lisanne de Vor**[1]*†, **Bruce van Dijk**[2]†, **Kok van Kessel**[1], **Jeffrey S Kavanaugh**[3], **Carla de Haas**[1], **Piet C Aerts**[1], **Marco C Viveen**[1], **Edwin C Boel**[1], **Ad C Fluit**[1], **Jakub M Kwiecinski**[3], **Gerard C Krijger**[4], **Ruud M Ramakers**[5,6,7], **Freek J Beekman**[5,6,7], **Ekaterina Dadachova**[8], **Marnix GEH Lam**[4], **H Charles Vogely**[2], **Bart CH van der Wal**[2], **Jos AG van Strijp**[1], **Alexander R Horswill**[3,9], **Harrie Weinans**[2,10], **Suzan HM Rooijakkers**[1]*

[1]Department of Medical Microbiology, University Medical Centre Utrecht, Utrecht, Netherlands; [2]Department of Orthopedics, University Medical Centre Utrecht, Utrecht, Netherlands; [3]Department of Immunology and Microbiology, University of Colorado School of Medicine, Aurora, United States; [4]Department of Radiology and Nuclear Medicine, University Medical Centre Utrecht, Utrecht, Netherlands; [5]MILabs B.V, Utrecht, Netherlands; [6]Department of Translational Neuroscience, Brain Center Rudolf Magnus, University Medical Center, Utrecht, Netherlands; [7]Department of Radiation Science and Technology, Delft University of Technology, Delft, Netherlands; [8]College of Pharmacy and Nutrition, University of Saskatchewan, Saskatoon, Canada; [9]Department of Veterans Affairs, Eastern Colorado Health Care System, Denver, United States; [10]Department of Biomechanical engineering, TU Delft, Delft, Netherlands

**\*For correspondence:**
l.devor-2@umcutrecht.nl (LdV);
s.h.m.rooijakkers@umcutrecht.nl (SHMR)

†These authors contributed equally to this work

**Abstract** Implant-associated *Staphylococcus aureus* infections are difficult to treat because of biofilm formation. Bacteria in a biofilm are often insensitive to antibiotics and host immunity. Monoclonal antibodies (mAbs) could provide an alternative approach to improve the diagnosis and potential treatment of biofilm-related infections. Here, we show that mAbs targeting common surface components of *S. aureus* can recognize clinically relevant biofilm types. The mAbs were also shown to bind a collection of clinical isolates derived from different biofilm-associated infections (endocarditis, prosthetic joint, catheter). We identify two groups of antibodies: one group that uniquely binds *S. aureus* in biofilm state and one that recognizes *S. aureus* in both biofilm and planktonic state. Furthermore, we show that a mAb recognizing wall teichoic acid (clone 4497) specifically localizes to a subcutaneously implanted pre-colonized catheter in mice. In conclusion, we demonstrate the capacity of several human mAbs to detect *S. aureus* biofilms in vitro and in vivo.

## Editor's evaluation

This article deals with the development of novel antibodies that are able to attach on both the planktonic and replicating phase of biofilm formed by *S. aureus*. The authors present evidence for the attachment ability of these antibodies using both in vitro experiments and animal experiments.

## Introduction

Implant-related infections are difficult to treat because of the ability of many bacterial species to form biofilm (*Arciola et al., 2018*). Biofilms are bacterial communities that adhere to abiotic surfaces (such as medical implants) using a self-made extracellular polymeric substance (EPS), consisting of proteins, polysaccharides, and extracellular DNA (*Otto, 2018*; *Schilcher and Horswill, 2020*). Bacteria in a biofilm are physically different from planktonic (free floating) bacteria and often more tolerant to antibiotics (*Beenken et al., 2004*). For instance, the EPS forms an important penetration barrier for many antimicrobial agents (*Otto, 2018*; *de Vor et al., 2020*). In addition, most antibiotics cannot kill bacteria in a biofilm because they are in a metabolically inactive state (*Resch et al., 2005*) and thus resistant to the antibiotics that act on active cellular processes (such as transcription/translation or cell wall formation; *Mah and O'Toole, 2001*). Another complication is that biofilm infections often occur in areas of the body that are not easily accessible for treatment without invasive surgical procedures. Consequently, treatment consists of long-term antibiotic regimens or replacement of the infected implant. Specific and noninvasive laboratory tests for early detection are not yet available and the diagnosis is often made only at advanced stages. This failure to detect biofilms adds further complications to effective diagnosis and treatment of these infections.

The human pathogen *Staphylococcus aureus* is the leading cause of healthcare-associated infections (*Lowy, 1998*; *Tong et al., 2015*). Today, 25% of healthcare-associated infections are related to medical implants such as heart valves, intravenous catheters, and prosthetic joints (*Magill et al., 2014*). *S. aureus* causes one-third of all implant-related infections in Europe and the United States (*Aggarwal et al., 2014*; *Arciola et al., 2005*) and is known for its ability to form biofilm (*Arciola et al., 2018*). Due to the absence of a vaccine and the emergence of methicillin-resistant *S. aureus* (MRSA), there is a clear need for diagnostic tools and alternative therapies for *S. aureus* biofilm infections.

Antibody-based biologicals could provide an alternative approach to improve the diagnosis and/or treatment of *S. aureus* biofilm-related infections. Monoclonal antibodies (mAb) may be exploited as vehicles to specifically bring anti-biofilm agents (such as radionuclides, enzymes, or photosensitizers) to the site of infection (*Tursi et al., 2020*; *Raafat et al., 2019*; *Lauderdale et al., 2010*; *Kaplan et al., 2012*; *Goodman et al., 2011*; *Brockson et al., 2014*; *Estellés et al., 2016*; *van Dijk et al., 2020*). Furthermore, radioactively labeled mAbs could be used for early diagnosis of biofilm-related infections. At present, only one mAb recognizing *S. aureus* biofilm has been identified. This F598 antibody recognizes poly-*N*-acetyl glucosamine (PNAG) (*Soliman et al., 2018*; *Soliman et al., 2020*) (also known as polysaccharide intercellular adhesion [PIA]; *Maira-Litrán et al., 2002*; *Mack et al., 1996*), a highly positively charged polysaccharide that was first recognized as a major EPS component of *S. aureus* biofilm. However, PNAG is not the only component of *S. aureus* biofilms. Recently, it has become clear that *S. aureus* may also use cell wall anchored proteins and eDNA to facilitate initial attachment and intercellular adhesion (*Cucarella et al., 2001*; *Corrigan et al., 2007*; *O'Neill et al., 2008*; *Moormeier et al., 2014*). In fact, deletion of the *icaADBC* locus (encoding PNAG) does not impair biofilm formation in multiple *S. aureus* strains (*Beenken et al., 2004*; *Moormeier et al., 2014*; *Boles et al., 2010*). These biofilms, referred to as PNAG-negative, are phenotypically different from PNAG-positive biofilm (*Mlynek et al., 2020*; *Rohde et al., 2007*; *O'Neill et al., 2007*; *Sugimoto et al., 2018*; *McCarthy et al., 2015*; *Fitzpatrick et al., 2005*). Because both types of biofilm occur in the clinic, we here focus on identifying mAbs that recognize PNAG-positive and PNAG-negative biofilm. In this study, we show that previously identified mAbs against staphylococcal surface structures can recognize both PNAG-negative and PNAG-positive *S. aureus* biofilms. Importantly, we show that some of these mAbs recognize *S. aureus* in both biofilm and planktonic state, which is crucial because release and dissemination of planktonic cells from biofilm-infected implants lead to life-threatening complications (*Lister and Horswill, 2014*). Finally, using SPECT/CT imaging, we show that radiolabeled mAbs have the potential to detect biofilm in vivo.

## Results

### Production of mAbs and validation of *S. aureus* biofilms

In order to study the reactivity of mAbs with *S. aureus* biofilms, we selected mAbs that were previously found to recognize surface components of planktonic *S. aureus* cells (*Raafat et al., 2019*; *Sause et al., 2016*). Specifically, we generated two antibodies recognizing cell wall teichoic acids (WTA) (4461-IgG

and 4497-IgG) (*Lehar et al., 2015*; *Fong et al., 2018*), one antibody against surface proteins of the SDR family (rF1-IgG) (*Hazenbos et al., 2013*), one antibody against clumping factor A (ClfA) (T1-2-IgG), and one antibody of which the exact target is yet unknown (CR5132-IgG) (*Figure 1A*). As a positive control, we generated F598-IgG against PNAG (*Soliman et al., 2018*). As negative controls, we produced one antibody recognizing the hapten dinitrophenol (DNP) (G2a-2-IgG) (*Gonzalez et al., 2003*) and one recognizing HIV protein gp120 (b12-IgG) (*Barbas et al., 1993*; *Saphire et al., 2001*). The variable heavy and light chain sequences of all antibodies were obtained from different scientific and patent publications (*Supplementary file 1*, *Kabat et al., 1984*) and cloned into expression vectors to produce full-length human IgG1 (kappa) antibodies in EXPI293F cells.

Since we were interested in the reactivity of these mAbs with both PNAG-positive and PNAG-negative biofilms, we selected two *S. aureus* to serve as models for these different biofilm phenotypes. We used Wood46 as a model strain for PNAG-positive biofilm because strain Wood46 is known to produce PNAG (*O'Brien et al., 2001*) and known for its low surface expression of IgG binding staphylococcal protein A (SpA). This is an advantage in antibody binding assays because nonspecific binding of the IgG1 Fc-domain to SpA complicates the detection of antibodies (*Amend et al., 1984*; *Balachandran et al., 2017b*). As a model strain for PNAG-negative biofilms, we used LAC (*Kennedy et al., 2008*; *Voyich et al., 2005*; *Montgomery et al., 2008*), a member of the USA300 lineage that has emerged as the common cause of healthcare-associated MRSA infections, including implant infections (*Seybold et al., 2006*; *Carrel et al., 2015*; *See et al., 2020*; *Haque et al., 2007*). Previous studies have demonstrated that LAC is capable of forming robust biofilm with no detectable PNAG (*Lauderdale et al., 2010*; *Moormeier et al., 2014*; *Mlynek et al., 2020*; *Lauderdale et al., 2009*; *Atwood et al., 2015*). Here, we used LACΔ*spa*Δ*sbi*, a mutant that lacks both SpA and a second immunoglobulin-binding protein (Sbi). To confirm the EPS composition of Wood46 and LACΔ*spa*Δ*sbi* biofilm, we treated biofilms with different EPS-degrading enzymes that degrade either PNAG (dispersin B [DspB]) or extracellular DNA (DNase I). As expected, LACΔ*spa*Δ*sbi* biofilm (*Figure 1—figure supplement 1A*) was sensitive to DNase I but not DspB while Wood46 biofilm (*Figure 1—figure supplement 1B*) was sensitive to DspB but not DNase I. The fact that Wood46 was insensitive to DNAse can be explained by shielding or DNA network stabilization by PNAG (*Mlynek et al., 2020*). At ultrastructural level, scanning electron microscopy (SEM) also verified the formation of phenotypically different biofilm by both strains (*Figure 1—figure supplement 1C and D*). Additionally, we verified that F598-IgG1, the only mAb in our panel that has been reported to bind biofilm (*Soliman et al., 2018*), indeed recognizes PNAG-positive biofilm of Wood46 (*Figure 1B*) but not PNAG-negative biofilm of LACΔ*spa*Δ*sbi* (*Figure 1C*).

## 4461-IgG1 and 4497-IgG1 against WTA recognize PNAG-positive and PNAG-negative *S. aureus* biofilm

Next, we tested the binding of other mAbs to *S. aureus* biofilms, starting with two well-defined antibodies recognizing WTA, the most abundant glycopolymer on the surface of *S. aureus* (*Brown et al., 2013*). mAbs 4461 and 4497 recognize different forms of WTA: while 4461 binds WTA with α-linked GlcNAc, 4497 recognizes β-linked GlcNAc (*Lehar et al., 2015*; *Fong et al., 2018*). The extent to which WTA is modified with GlcNAc depends both on the presence of genes encoding enzymes responsible for α- or β-glycosylation (*Li et al., 2015*) and the expression of these genes based on environmental conditions (*Mistretta et al., 2019*). First, we studied binding of 4461-IgG1 and 4497-IgG1 to exponential planktonic cultures of Wood46 and LACΔ*spa*Δ*sbi* (*Figure 2A*). In line with the fact that Wood46 is negative for the enzyme responsible for α-GlcNAc glycosylation of WTA (TarM; *Mistretta et al., 2019*), we observed no binding of 4461-IgG1 to planktonic Wood46. In contrast, 4461-IgG1 bound strongly to planktonic LACΔ*spa*Δ*sbi*. For 4497-IgG, we observed that 4497-IgG1 bound strongly to planktonic Wood46 cells but very weakly to planktonic LACΔ*spa*Δ*sbi* (*Figure 2A*).

Upon studying binding of WTA-specific antibodies to biofilms, we observed that 4497-IgG1 strongly bound to PNAG-positive biofilm formed by Wood46 (*Figure 2B*). While F598-IgG1 exclusively binds PNAG-positive biofilms but not planktonic *S. aureus* (*Figure 2—figure supplement 1*), 4497-IgG1 can bind *S. aureus* Wood46 in both planktonic and biofilm states (Figure 2A and B). This is important because in the biofilm life cycle planktonic cells can be released from a biofilm and disseminate to other locations in the body (*Lister and Horswill, 2014*). Apart from recognizing PNAG-positive biofilms, 4497-IgG1 also bound the PNAG-negative biofilm formed by LACΔ*spa*Δ*sbi* (*Figure 2B*). This

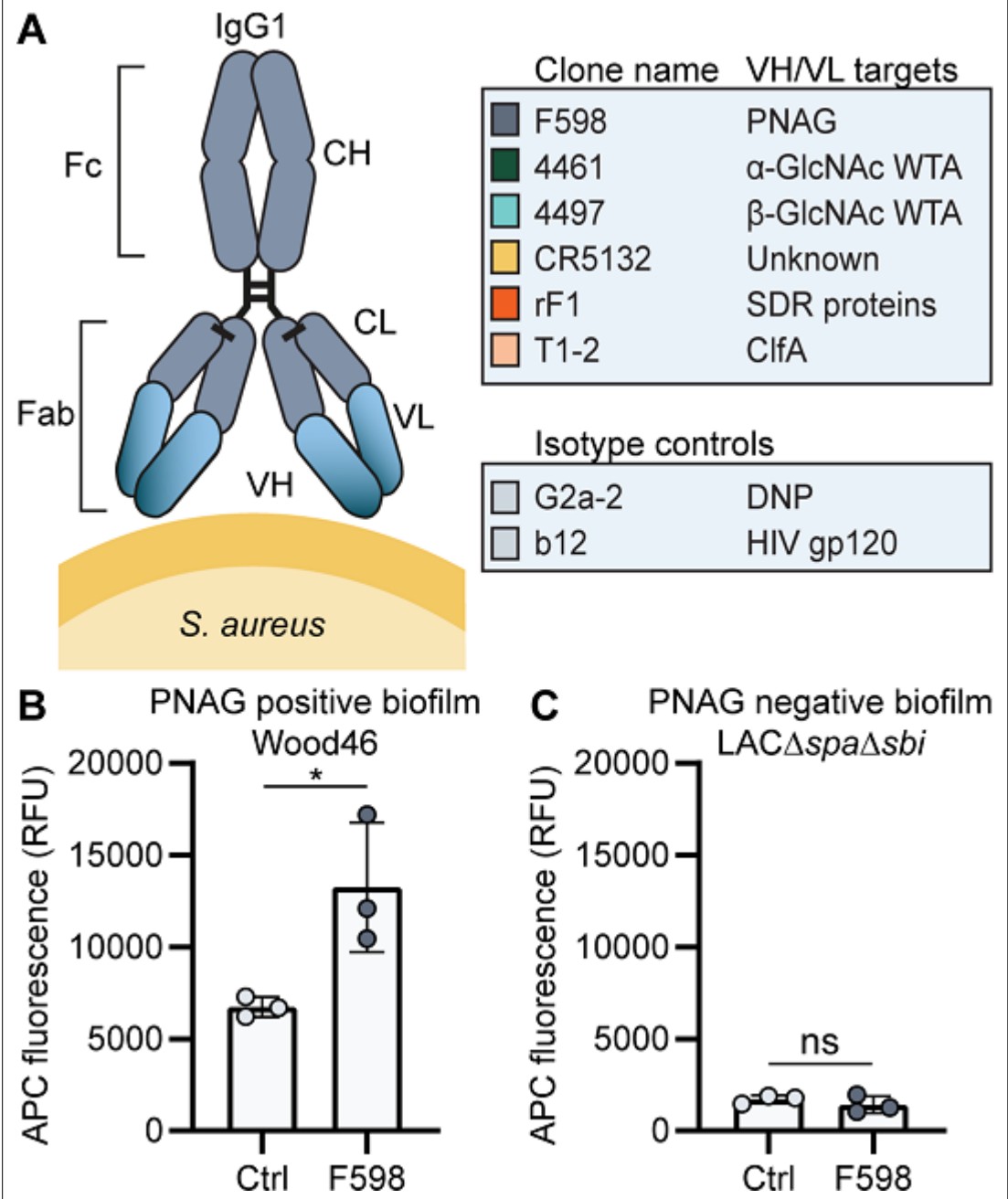

**Figure 1.** Production of monoclonal antibodies (mAbs) and validation of biofilm. (**A**) Human IgG1 antibodies are large (150 kDa) proteins, consisting of two functional domains. The fragment antigen binding (Fab) region confers antigen specificity, while the crystallizable fragment (Fc) region drives interactions with the immune system. Each IgG1 is composed of two identical heavy chains and two identical light chains, which all consist of a constant (CH, CL) and a variable (VH, VL) domain. A panel of six human IgG1 mAbs that recognize polysaccharide and protein components on the cell surface of *S. aureus* and two nonspecific isotype controls was produced. Variable heavy (VH) and light (VL) chain sequences obtained from different scientific and patent publications were cloned in homemade expression vectors containing human heavy chain (HC) and light chain (LC) constant regions, respectively. (**B, C**) Biofilms of Wood46 (**B**) and LAC*ΔspaΔsbi* (**C**) were grown for 24 hr and incubated with 66 nM F598-IgG1 or ctrl-IgG1 (G2a-2). mAb binding was detected using APC-labeled anti-human IgG antibodies and a plate reader and plotted as fluorescence intensity per well. Data represent mean + SD of three independent experiments. A ratio paired t-test was performed to test for differences in antibody binding versus control and displayed only when significant as *p≤0.05, **p≤0.01, ***p≤0.001, or ****p≤0.0001. Exact p-values are displayed in *Supplementary file 2*.

The online version of this article includes the following figure supplement(s) for figure 1:

**Figure supplement 1.** *S. aureus* strains LAC and Wood46 form different types of biofilm.

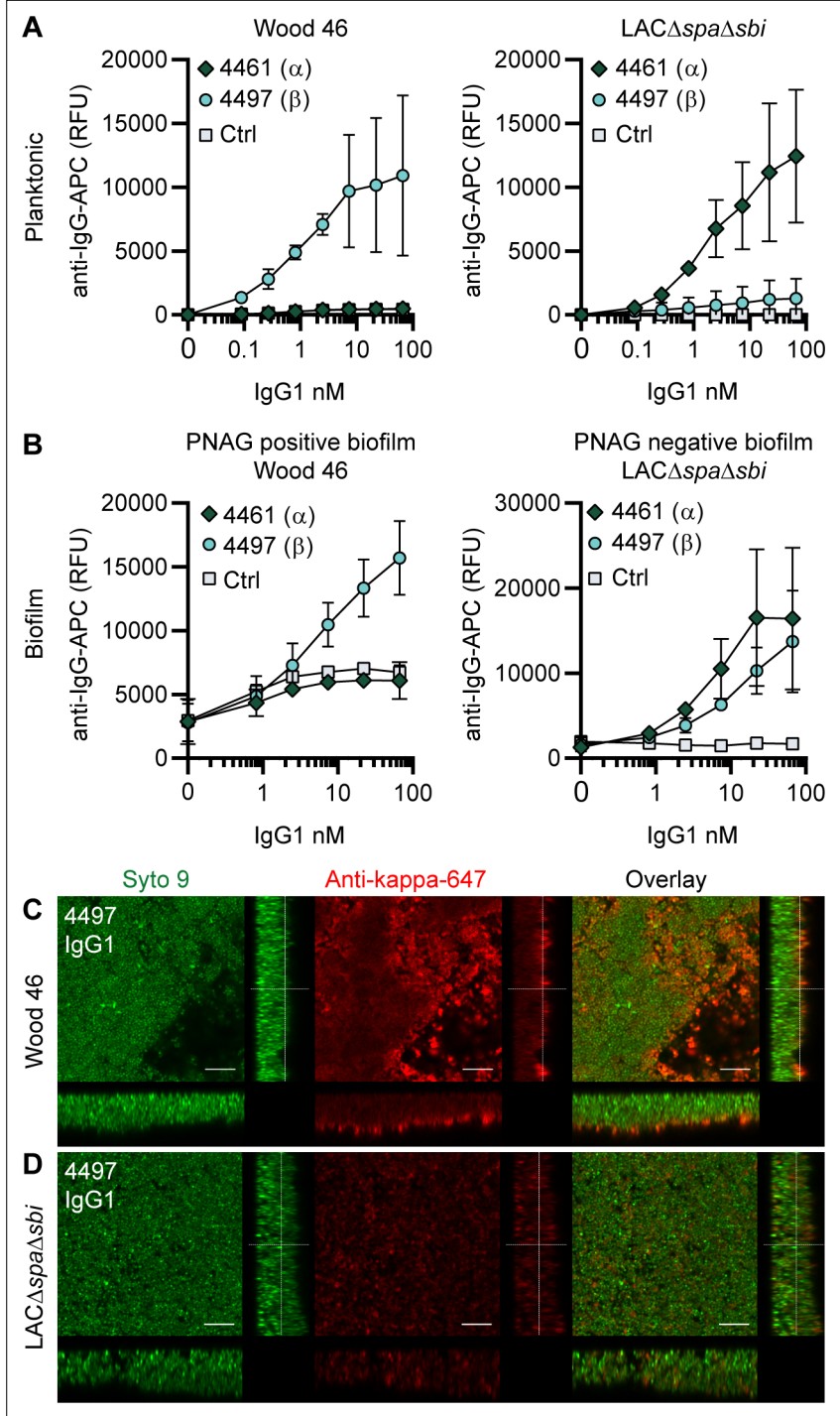

**Figure 2.** IgG1 monoclonal antibodies (mAbs) against wall teichoic acid (WTA) bind *S. aureus* in planktonic and biofilm mode. (**A**) Planktonic bacteria of Wood46 (left) and LAC*ΔspaΔsbi* (right) were grown to exponential phase and incubated with a concentration range of 4461-IgG1 or 4497-IgG1. mAb binding was detected using APC-labeled anti-human IgG antibodies and flow cytometry and plotted as geoMFI of the bacterial population. (**B**) Biofilms of Wood46 (left) and LAC*ΔspaΔsbi* (right) were grown for 24 hr and incubated with a concentration range of 4461-IgG1 or 4497-IgG1. mAb binding was detected using APC-labeled anti-human IgG antibodies and a plate reader and plotted as fluorescence intensity per well. Data represent mean + SD of three independent experiments. (**C, D**) Biofilm was grown for 24 hr and incubated with 66 nM IgG1 mAb. Bacteria were visualized by Syto9 (green), and mAb binding was detected by staining with Alexa Fluor 647-conjugated goat-anti-human-kappa F(ab')2 antibody (red). Orthogonal views are representative for a total of three Z-stacks per condition and at least

*Figure 2 continued on next page*

*Figure 2 continued*

two independent experiments. Scale bars: 10 μm.

The online version of this article includes the following figure supplement(s) for figure 2:

**Figure supplement 1.** F598-IgG1 binds poly-N-acetyl glucosamine (PNAG)-dependent biofilms specifically.

**Figure supplement 2.** Background control monoclonal antibody (mAb) binding to Wood46 biofilm due to incorporation of secreted SpA in biofilm.

**Figure supplement 3.** Orthogonal views of poly-N-acetyl glucosamine (PNAG)-negative biofilm incubated with IgG1 monoclonal antibodies (mAbs).

**Figure supplement 4.** Orthogonal views of poly-N-acetyl glucosamine (PNAG)-positive biofilm incubated with IgG1 monoclonal antibodies (mAbs).

---

is remarkable because 4497-IgG1 did not potently bind planktonic LAC*ΔspaΔsbi* (*Figure 2A*). Finally, we observe that also 4461-IgG1 effectively recognizes PNAG-negative biofilms. In all, these data identify mAbs against WTA as potent binders of PNAG-positive (4497) and PNAG-negative (4461 and 4497) biofilms.

Because we observed background binding of control IgG1 to Wood46 biofilm compared to planktonic Wood46 (*Figure 2—figure supplement 1*), we wondered whether this could be explained by secreted SpA being incorporated in the biofilm matrix as Wood46 is unable to link secreted SpA to the surface due to a sortase defect (*Balachandran et al., 2017a*). To test this hypothesis, we performed a binding assay on planktonic versus biofilm Wood46 using nonspecific IgG1, nonspecific IgG3 (which is unable to bind to SpA via the Fc-domain *Jendeberg et al., 1997*), and anti-SpA-IgG3 (binding SpA via the Fab-domain but not the Fc-domain). Here, we observed high binding of anti-SpA-IgG3 to Wood46 biofilm (*Figure 2—figure supplement 2A*) but not planktonic (*Figure 2—figure supplement 2B*) bacteria. Thus, SpA is incorporated in Wood46 biofilm but is washed away in the planktonic binding assay.

Using confocal microscopy as an independent method, we confirmed binding of anti-WTA mAbs to in vitro biofilm. Biofilm was cultured in chambered microscopy slides and incubated with IgG1 mAbs or isotype controls (*Figure 2—figure supplement 3*, *Figure 2—figure supplement 4*). Bound mAbs were detected by using AF647-labeled anti-human-kappa-antibodies; bacteria were visualized using DNA dye Syto9. A total of three Z-stacks were acquired at random locations in each chamber of the slide. Z-stacks were visualized as orthogonal views. Using this technique, we visualized binding of 4497-IgG1 to PNAG-positive (*Figure 2C*) and PNAG negative biofilm (*Figure 2D*) and binding of 4461-IgG1 to PNAG-negative biofilm (*Figure 2—figure supplement 3*). Importantly, isotype controls showed no binding (*Figure 2—figure supplement 3*, *Figure 2—figure supplement 4*). In conclusion, we show that mAbs recognizing polysaccharides WTA α-GlcNAc and WTA β-GlcNAc are able to bind their targets when bacteria are growing in biofilm mode.

## CR5132-IgG1 discriminates between planktonic bacteria and biofilm

mAb CR5132 was discovered through phage display libraries from human memory B cells (US 2012/0141493 A1) and was selected for binding to staphylococcal colonies scraped from plates. Since such colonies more closely resemble a surface attached biofilm than free-floating cells (*Serra et al., 2015*), we were curious whether this mAb could recognize biofilm. Intriguingly, CR5132-IgG1 showed almost no detectable binding to exponential planktonic LAC*ΔspaΔsbi* or Wood46 (*Figure 3A*), but it bound strongly to both PNAG-negative and PNAG-positive biofilms formed by these strains (*Figure 3B*). Confocal microscopy confirmed CR5132-IgG1 binding to PNAG-positive (*Figure 3C*) and PNAG-negative biofilms (*Figure 3D*). The ability of CR5132-IgG1 to target both types of *S. aureus* biofilms and to discriminate between planktonic bacteria and biofilm makes CR5132 a unique and interesting mAb. Because of the interesting binding phenotype of CR5132-IgG1, we performed experiments to identify its target. LTA was originally identified as one of the targets of CR5132 (US 2012/0141493 A1), but the quality of commercial LTA preparations varies greatly and often contains other components (*Morath et al., 2002*; *Morath et al., 2005*). Therefore, we first tested CR5132-IgG1 binding to *S. aureus* purified cell wall components LTA and peptidoglycan coated on ELISA plates. As a positive control, we used the established A120-IgG1, which is known to bind to LTA (EP2027155A2). Interestingly, we could not detect CR5132-IgG1 binding to LTA (*Figure 3—figure supplement 1A*) or

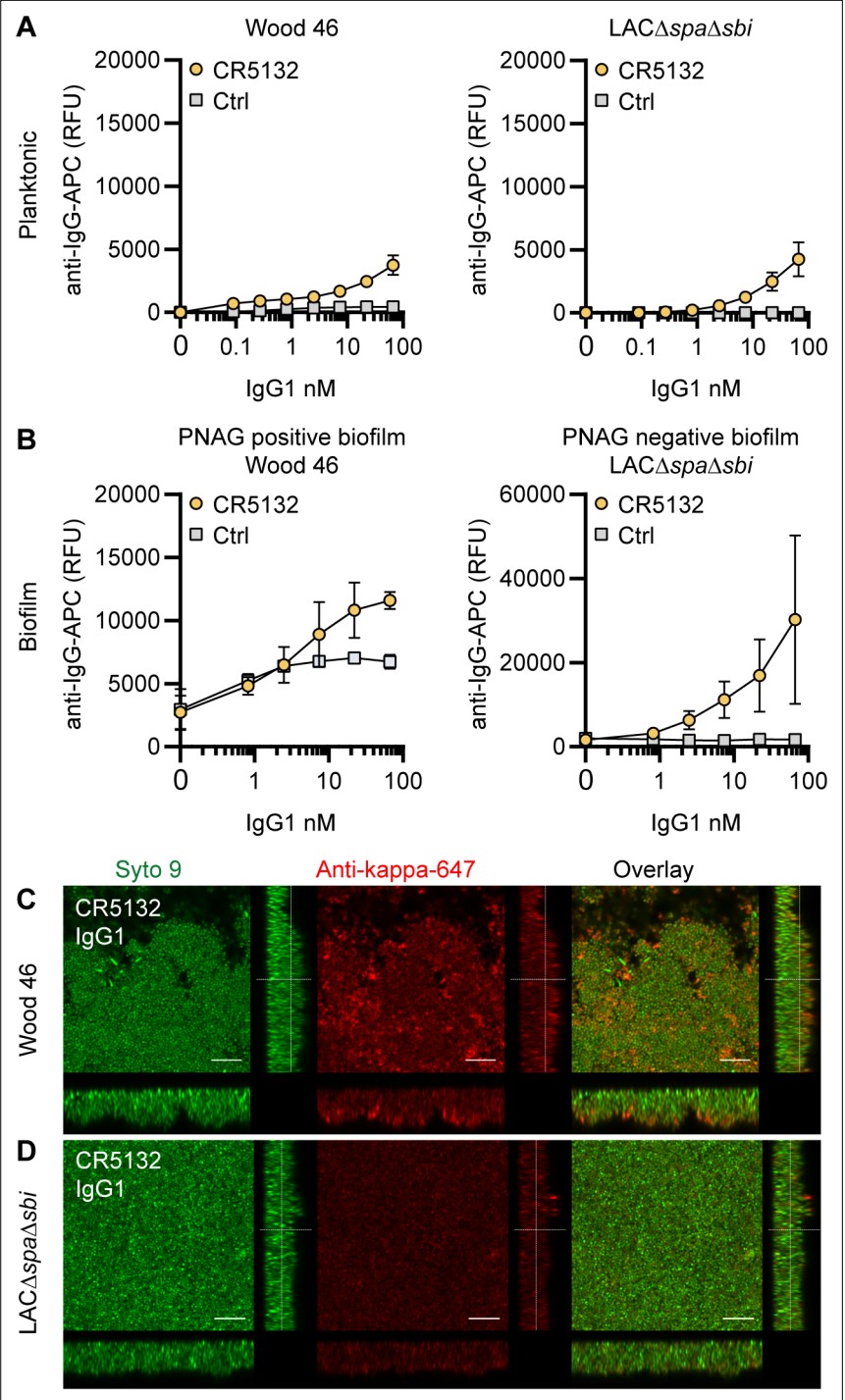

**Figure 3.** CR5132-IgG1 discriminates between planktonic bacteria and biofilm. (**A**) Planktonic bacteria of Wood46 (left) and LAC*ΔspaΔsbi* (right) were grown to exponential phase and incubated with a concentration range of CR5132-IgG1. Monoclonal antibody (mAb) binding was detected using APC-labeled anti-human IgG antibodies and flow cytometry and plotted as geoMFI of the bacterial population. (**B**) Biofilms of Wood46 (left) and LAC*ΔspaΔsbi* (right) were grown for 24 hr and incubated with a concentration range of CR5132-IgG1. mAb binding was detected using APC-labeled anti-human IgG antibodies and a plate reader and plotted as fluorescence intensity per well. Data represent mean + SD of at least three independent experiments. (**C, D**) Biofilm was grown for 24hr and incubated with 66 nM IgG1 mAb. Bacteria were visualized by Syto9 (green), and mAb binding was detected by staining with Alexa Fluor 647-conjugated goat-anti-human-kappa F(ab')₂ antibody (red). Orthogonal views are representative for a total of three Z-stacks per condition and at least two independent experiments.

*Figure 3 continued on next page*

*Figure 3 continued*

Scale bars: 10 µm.

The online version of this article includes the following figure supplement(s) for figure 3:

**Figure supplement 1.** Target identification of CR5132.

peptidoglycan (*Figure 3—figure supplement 1B*), while A120-IgG1 showed detectable binding to LTA. Next, we tested CR5132-IgG1 binding to pure α-GlcNAc or β-GlcNAc WTA structures. To do this, we used magnetic beads that were artificially coated with the WTA backbone and then glycosylated by recombinant TarM, TarS, or TarP, resulting in pure β 1,4-GlcNAc, β 1,3- GlcNAc, or α 1,4-GlcNAc WTA structures in their natural conformation on a surface (*van Dalen et al., 2019*). This way, we identified WTA β-GlcNAc instead of LTA as one of the targets of CR5132 (*Figure 3—figure supplement 1C*).

## RF1-IgG1 against the SDR protein family binds *S. aureus* in planktonic and biofilm form

Finally, we tested whether mAbs recognizing proteins on the staphylococcal cell surface are able to bind *S. aureus* biofilm. mAb rF1 recognizes the SDR family of proteins, which is characterized by

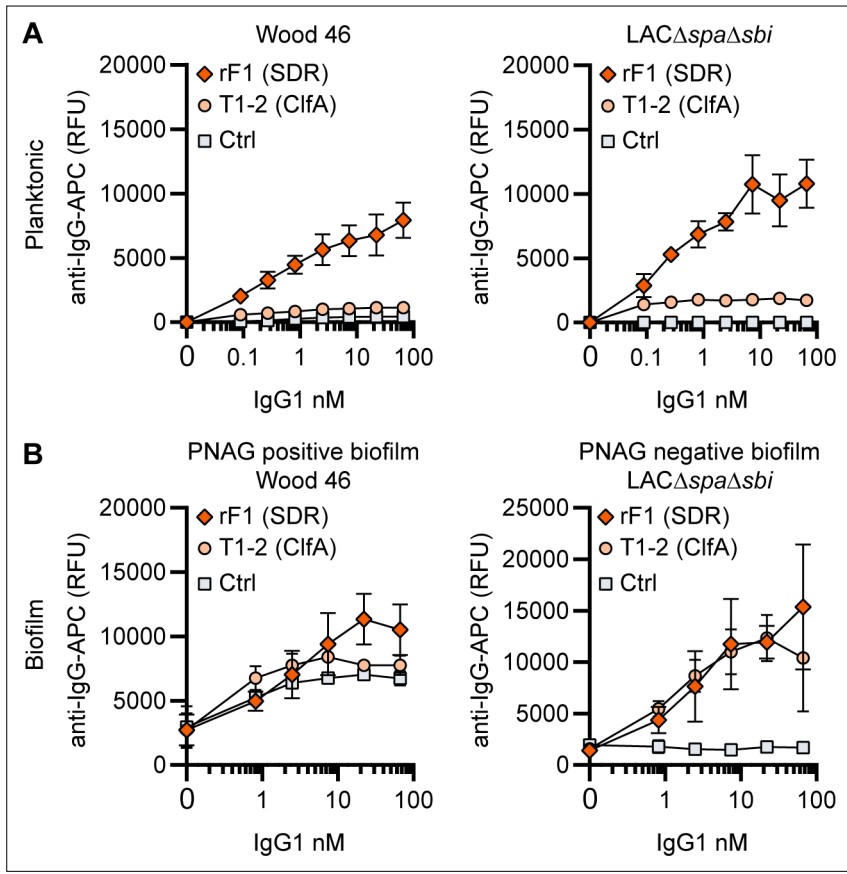

**Figure 4.** IgG1 monoclonal antibodies (mAbs) against protein components bind planktonic bacteria as well as biofilm. (**A**) Planktonic bacteria of Wood46 (left) and LAC*ΔspaΔsbi* (right) were grown to exponential phase and incubated with a concentration range of rF1-IgG1 or T1-2-IgG1. mAb binding was detected using APC-labeled anti-human IgG antibodies and flow cytometry and plotted as geoMFI of the bacterial population. (**B**) Biofilms of Wood46 (left) and LAC*ΔspaΔsbi* (right) were grown for 24 hr and incubated with a concentration range of rF1-IgG1 or T1-2-IgG1. mAb binding was detected using APC-labeled anti-human IgG antibodies and a plate reader and plotted as fluorescence intensity per well. Data represent mean + SD of three independent experiments.

The online version of this article includes the following figure supplement(s) for figure 4:

**Figure supplement 1.** Binding of the monoclonal antibody (mAb) panel to stationary phase planktonic cultures.

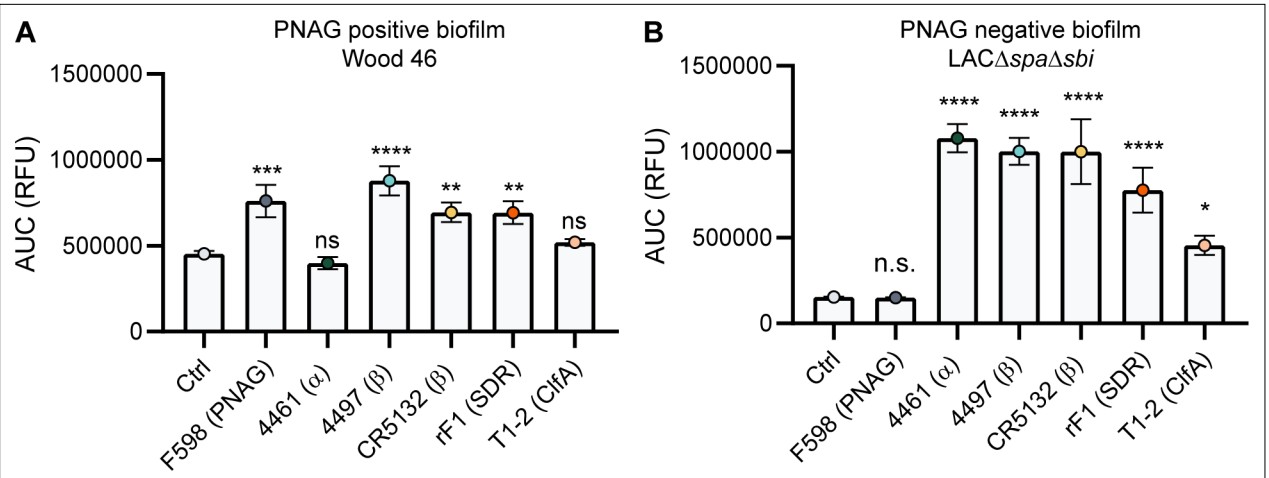

**Figure 5.** Comparative binding of IgG1 monoclonal antibodies (mAbs) to *S. aureus* biofilm. Biofilms of Wood46 (**A**) and LAC*ΔspaΔsbi* (**B**) were grown for 24 hr and incubated with a concentration range of IgG1 mAbs. mAb binding was detected using APC-labeled anti-human IgG antibodies and a plate reader. Data are expressed as area under the curve (AUC) of the binding curve (mean + SD) of three independent experiments. One-way ANOVA followed by Dunnett test was performed to test for differences in antibody binding versus control and displayed only when significant as *p≤0.05, **p≤0.01, ***p≤0.001, or ****p≤0.0001. Exact p-values are displayed in *Supplementary file 2*.

The online version of this article includes the following figure supplement(s) for figure 5:

**Figure supplement 1.** Comparative binding of IgG1 monoclonal antibodies (mAbs) to planktonic bacteria.

**Figure supplement 2.** Mean total fluorescence per Z-stack corresponds to plate reader data.

**Figure supplement 3.** Binding in the presence of pooled serum IgG.

**Figure supplement 4.** Binding of IgG3 monoclonal antibodies (mAbs) to planktonic and biofilm LAC wild type.

a large stretch of serine-aspartate dipeptide repeats (SDR) and includes *S. aureus* ClfA, clumping factor B (ClfB), and SDR proteins C, D, and E and three additional SDR proteins from *Staphylococcus epidermidis* (*Josefsson et al., 1998*). mab rF1 recognizes glycosylated SDR repeats that are present in all members of this protein family. Additionally, the well-described mAb T1-2 recognizes SDR family member ClfA (*Ganesh et al., 2016*; *Domanski et al., 2005*). We confirmed effective binding of rF1-IgG1 to exponential planktonic cultures of Wood46 and LAC*ΔspaΔsbi* (*Figure 4A*). In addition, both PNAG-positive and PNAG-negative biofilms formed by these strains were bound by rF1-IgG1 (*Figure 4B*). T1-2-IgG1 binding to planktonic bacteria was only detectable in stationary LAC*ΔspaΔsbi* cultures (*Figure 4—figure supplement 1*) and not in exponential cultures (*Figure 4A*) because ClfA is known to be expressed in the stationary phase (*Wolz et al., 2002*). Furthermore, effective binding of T1-2-IgG1 to LAC*ΔspaΔsbi* PNAG-negative biofilm was detected (*Figure 4B*). In contrast, we could not detect T1-2-IgG1 binding to Wood46 PNAG-positive biofilm (*Figure 4B*). This difference in binding might be explained by a greater abundance of ClfA in PNAG-negative biofilm than PNAG-positive biofilm or PNAG shielding ClfA from T1-2-IgG1 binding. In conclusion, we show that rF1-IgG1 and T1-2-IgG1 bind surface proteins on planktonic bacteria as well as biofilm formed by these bacteria. This means that besides *S. aureus* surface polysaccharides, surface proteins in a biofilm can also be recognized by mAbs.

## Comparative binding of mAbs to *S. aureus* biofilm

A direct comparison of all biofilm-binding mAbs revealed 4497-IgG1 as the best binder to PNAG-positive biofilm (*Figure 5A*) and CR5132 as the best binder to PNAG-negative biofilm (*Figure 5B*). Furthermore, all mAbs that bind to exponential planktonic bacteria (*Figure 5—figure supplement 1*) were able to bind biofilm (*Figure 5*) formed by that strain. Additionally, some mAbs, that is, F598-IgG1 (anti-PNAG) and CR5132-IgG1 (anti-β-GlcNAc WTA), showed enhanced binding to biofilm compared to planktonic bacteria. Thus, we can identify two classes of mAbs: one class recognizing both planktonic bacteria and biofilm, and one class recognizing biofilm only (*Table 1*). Importantly, the mean AF647 fluorescence levels of Z-stacks acquired with the microscope (*Figure 5—figure supplement 2*) corresponded to our plate reader data (*Figure 5*). As most humans possess antibodies against *S.*

**Table 1.** Monoclonal antibody (mAb) binding to biofilm and planktonic bacteria.
Significant binding (p<0.05) of IgG1 mAbs compared to control IgG1 s indicated with '+,' weak binding (p<0.05, p<0.99) is indicated with '+/-,' and no significant binding (p>0.99) is indicated with '−.'.

| Clone | Target | Biofilm | | Planktonic | |
| --- | --- | --- | --- | --- | --- |
| | | PNAG (+) | PNAG (-) | Wood46 | LAC ΔspaΔsbi |
| F598 | PNAG | + | – | +/- | – |
| 4461 | WTA(α) | +/- | + | – | + |
| 4497 | WTA(β) | + | + | + | +/- |
| CR5132 | WTA(β) | + | + | +/- | + |
| rF1 | SDR proteins | + | + | + | + |
| T1-2 | ClfA | +/- | + | +/- | +/- |

PNAG, poly-*N*-acetyl glucosamine; WTA, wall teichoic acid; SDR, serine-aspartate dipeptide repeats; ClfA, clumping factor B.

*aureus*, we wondered whether preexisting antibodies might compete with the IgG1 mAbs for binding to epitopes. To test this possibility, biofilm cultures were incubated with AF647-labeled mAbs in the presence of excess IgG (mAb:IgG ratio 1:25) isolated from pooled human serum. This ratio was based on ongoing clinical trials for mAb therapy for *S. aureus* infections (NCT02296320), where 2 g and 5 g is administered to patients, reaching a 1:25 mAb:natural IgG ratio in the human circulation. Despite the excess IgG, the AF647-labeled mAbs retained, on average, approximately 60% of the fluorescence they had in the absence of IgG (*Figure 5—figure supplement 3*). This indicates that the mAbs are able to recognize *S. aureus* biofilm in the presence of preexisting antibodies.

## The majority of mAbs recognize PNAG-positive and PNAG-negative biofilm formed by clinical isolates from biofilm-associated infections

Because clinical *S. aureus* isolates express SpA, we wanted to test mAb binding in the presence of this surface protein. To rule out nonspecific binding, we produced all mAbs in the IgG3 subclass, which is unable to bind SpA via the Fc domain (*Jendeberg et al., 1997*). Then, we compared our data acquired with LACΔspaΔsbi to the LAC WT strain. Binding of IgG3 mAbs to LAC WT in planktonic (*Figure 5—figure supplement 4A*) and biofilm (*Figure 5—figure supplement 4B*) was comparable to binding of IgG1 mAbs to planktonic (*Figure 5—figure supplement 1B*) and biofilm (*Figure 5B*) LACΔspaΔsbi. Interestingly, we observed binding of 4497-IgG3 to LAC WT (*Figure 5—figure supplement 4*), suggesting that knocking out *spa* and *sbi* altered the WTA glycosylation pattern.

Next, we wanted to test if our data acquired on two model bacterial strains translated to clinical isolates from patients with biofilm-related infections. In literature, no correlation between *S. aureus* biofilm phenotypes (PNAG-positive and PNAG-negative) and the source of clinical biofilm infections has been described. Therefore, we collected a variety of *S. aureus* isolates from endocarditis (n = 4), prosthetic joint infections (PJIs) (n = 16), and catheter tip infections (n = 25). First, we determined whether these clinical isolates produced PNAG-positive or PNAG-negative biofilm by using a crystal violet assay and staining with F598-IgG3 (unable to bind SpA). We could detect significant F598-IgG3 binding to 1/4 endocarditis isolates, 5/25 catheter tip isolates, and 6/16 PJI isolates (*Figure 6A*). This indicates that production of PNAG is not a hallmark of one specific source of biofilm-related infections and that approximately one-third of isolates form PNAG-positive biofilm in vitro. This observation also underlines the importance of identifying mAbs that recognize both types of biofilm. As expected, there was a high variation in the amount of biofilm formation and the amount of PNAG produced (*Figure 6B*). These data show that our model bacterial strains Wood46 and LACΔspaΔsbi represent the different types of biofilm that is formed by clinical isolates. Next, we tested binding of the other anti-*S. aureus* IgG3 mAbs to six PNAG-positive clinical isolates and six PNAG-negative clinical isolates that were good biofilm formers (*Figure 6C*). Most importantly, we found that 4/6 mAbs (4497, CR5132, rF1, T1-2) recognize PNAG-positive and PNAG-negative biofilm formed by all clinical isolates. Furthermore, we found that mAb 4461 (against α-GlcNAc WTA) recognizes 4 out of total 12

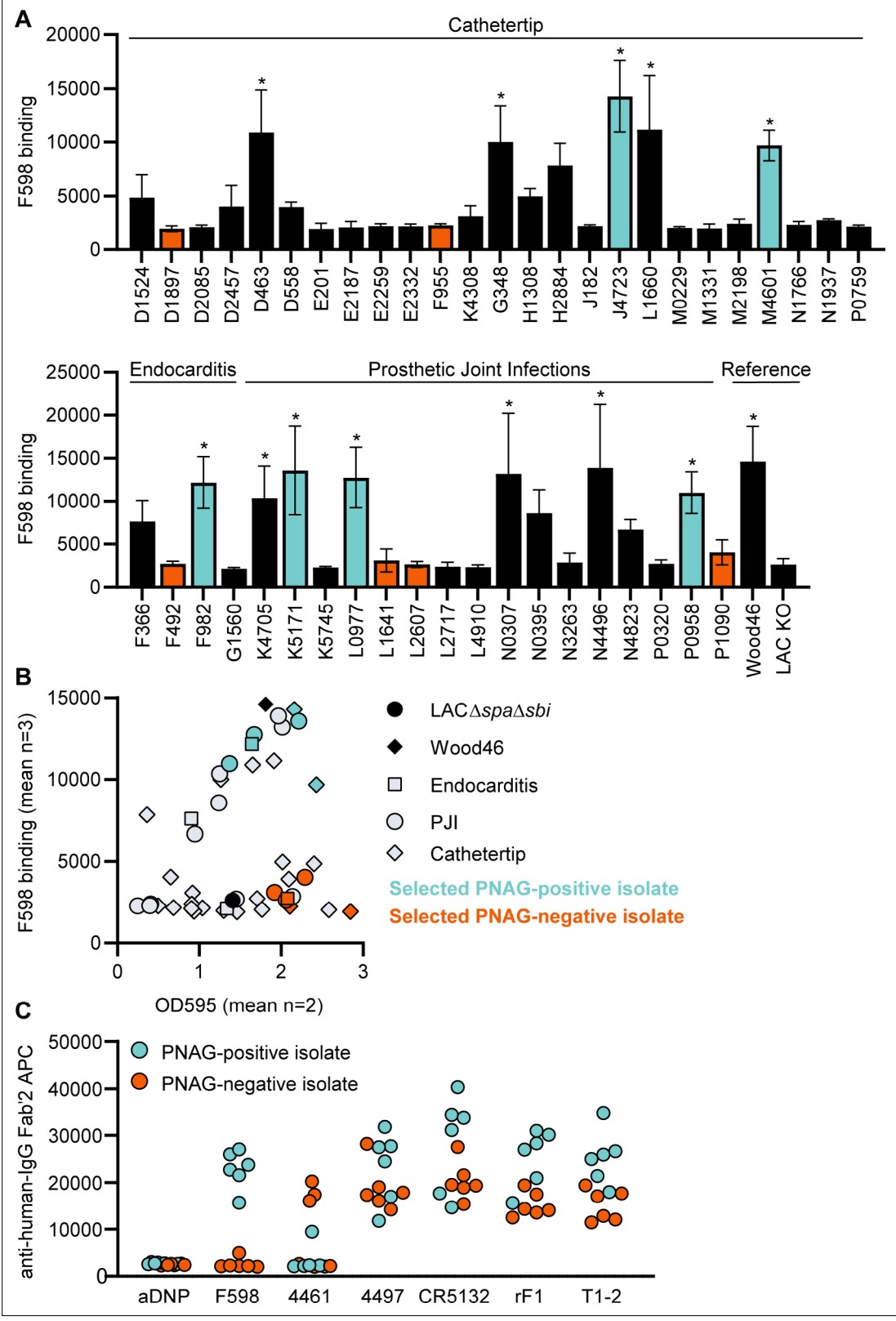

**Figure 6.** Binding of monoclonal antibodies (mAbs) to *S. aureus* clinical isolate biofilm. (**A**) Biofilm of clinical isolates derived from catheter tip, endocarditis, and prosthetic joint infections (PJIs) was grown for 24 hr and incubated with 33 nM F598-IgG3. mAb binding was detected using APC-labeled anti-human IgG antibodies and a plate reader. (**B**) Scatter plot of F598-IgG3 binding to isolates and biofilm adherent biomass measured by crystal violet staining after mAb binding assay. Isolates selected for (**C**) are indicated. (**C**) Biofilms of clinical isolates was grown for 24 hr and incubated

*Figure 6 continued on next page*

*Figure 6 continued*

with 33 nM IgG3 mAbs. mAb binding was detected using APC-labeled anti-human IgG antibodies and a plate reader. Data (**A**) represent mean + SD of three independent experiments. One-way ANOVA followed by Dunnett test was performed to test for differences in antibody binding versus LAC KO and displayed only when significant as *. Exact p-values are displayed in *Supplementary file 2*. Data (**B**) represent mean two independent experiments.

clinical isolates, in line with literature describing 35.7% of clinical isolates being TarM positive (*Winstel et al., 2015*).

## Indium-111 labeled 4497-IgG1 localizes to subcutaneously implanted pre-colonized catheter in mice

Lastly, we studied whether mAbs against *S. aureus* biofilm could be used to localize in vivo to a subcutaneous implant pre-colonized with biofilm. Mice received a 5 mm catheter that was pre-colonized with *S. aureus* biofilm in one flank. As an internal control, a sterile catheter was inserted into the other flank. Pre-colonized catheters were generated by incubating catheters with *S. aureus* USA300 LAC (AH4802; *Miller et al., 2019*) for 48 hr. Bacterial loads on the catheters before implantation were approximately $4.5 \times 10^7$ CFU (*Figure 7—figure supplement 1*). We selected 4497-IgG1 (against β-GlcNAc WTA) because it potently binds to LAC biofilm in vitro (*Figure 5*). To detect antibody localization in the mouse body, we radiolabeled 4497-IgG1 with indium-111 ($^{111}$In). Two days after implantation of the catheters, mice were injected intravenously with $^{111}$In-labeled 4497-IgG1 and distribution of the radiolabel was visualized with total-body SPECT-CT scans at 24, 72, and 120 hr after injection. Maximum intensity projections of SPECT/CT scans showed typical distribution patterns for IgG distribution in mice (*Allen et al., 2018*; *Yip et al., 2014*). At 24 hr, activity was detected in blood-rich organs such as heart, lungs, and liver (*Figure 7A*, *Figure 7—figure supplement 2*, *Figure 7—animation 1*). In line with literature describing 2–3 days half-life of human IgG1 in mice (*Allen et al., 2019*), antibodies were cleared from the circulation and blood-rich organs over time, while the specific activity of radiolabeled 4497-IgG1 around pre-colonized implants remained. Remaining activity that was detected at incision sites of the pre-colonized catheters was likely explained by nonspecific accumulation of antibodies at inflammatory sites.

To quantify the amount of antibody accumulating at pre-colonized and sterile implants, a volume of interest was drawn manually around the implants visible on SPECT-CT. The activity measured in the volume of interest was quantified as a percentage of the total body activity (*Figure 7B*). At all time points, 4497-IgG1 accumulated selectively at the pre-colonized catheter with a mean of 7.7% (24 hr), 8.1% (72 hr), and 6.4% (120 hr) of the total body activity in the region of interest around the pre-colonized implant compared to 1.1% (24 hr), 0.7% (72 hr), and 0.2% (120 hr) around the sterile implant. At each time point, we could detect a significant difference in 4497-IgG1 localization to pre-colonized implants compared to sterile implants. The same results were found in a similar pilot experiment with one mouse and less mAbs administered (*Figure 7—figure supplement 4*). At the end point (120 hr), thus 5 days after implantation of the catheter, CFU counts on implants (n = 3) were determined and a mean of $\sim 1.1 \times 10^6$ CFU were recovered from pre-colonized implants, whereas no bacteria were recovered from sterile controls (*Figure 7—figure supplement 1*). Interestingly, when a higher bacterial burden was recovered from a pre-colonized implant (n = 3) at the end point (*Figure 7—figure supplement 1*, each shape is one mouse), a higher 4497-IgG1 activity was measured at the implant (*Figure 7B*, 120 hr, see corresponding shapes), suggesting that a larger infection recruits more specific antibodies.

We used an SpA-expressing LAC USA300 strain in vivo because *S. aureus* clinical isolates express SpA. To control for nonspecific binding of mAbs via the IgG1 Fc-tail, we used nonspecific $^{111}$In-labeled palivizumab (an antiviral IgG1) in a different set of mice. In two out of four mice, we saw increased $^{111}$In activity at the colonized implant compared to the sterile implant. $^{111}$In-labeled palivizumab was detected at pre-colonized catheters with a mean of 5.0% (24 hr), 5.2% (72 hr), and 2.9% (120 hr) of the total body radiolabel activity and at sterile catheters with 1.4% (24 hr), 0.4% (72 hr), and 0.2% (120 hr) (*Figure 7—figure supplement 3*). Because the mean $^{111}$In-labeled 4497-IgG1 localization to colonized implants was higher than the mean $^{111}$In-labeled palivizumab localization at each time point (6.4% vs. 2.9% at 120 hr), localization of $^{111}$In-labeled 4497-IgG1 is likely a combination of specific and nonspecific binding at the colonized implant.

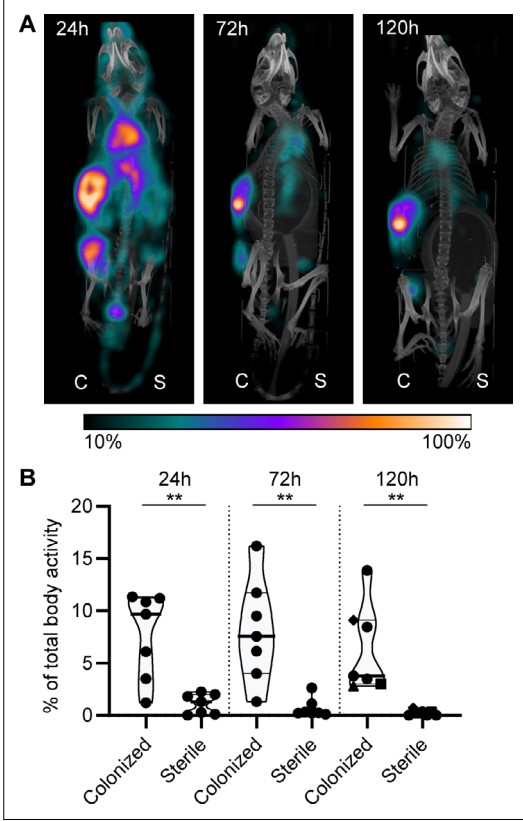

**Figure 7.** Localization of [¹¹¹In]In-4497-IgG1 to a subcutaneous implant pre-colonized with biofilm. Two days after implantation, mice were injected with 7.5 MBq [¹¹¹In]In-4497-IgG1 (n = 7) and imaged at 24 hr, 72 hr, and 120 hr after injection. (**A**) Maximum intensity projection (corrected for decay) of a mouse subcutaneously bearing pre-colonized (C; left flank) and sterile (S; right flank) catheter. Additional scans can be seen in the supplementary information (*Figure 7—figure supplement 2*). (**B**) The activity detected in regions of interests was expressed as a percentage of total body activity. Each data point represents one mouse. A two-tailed paired t-test was performed to test for differences in activity in sterile versus colonized implants displayed as *p≤0.05, **p≤0.01, ***p≤0.001, or ****p≤0.0001. Exact p-values are displayed in *Supplementary file 2*.

The online version of this article includes the following video and figure supplement(s) for figure 7:

**Figure supplement 1.** CFU count before implantation and after implantation.

**Figure supplement 2.** Localization of [¹¹¹In]In-4497-IgG1 to subcutaneous implant pre-colonized with biofilm in a mouse model.

**Figure supplement 3.** Localization of [¹¹¹In]In-palivizumab to a subcutaneous implant pre-colonized with biofilm.

**Figure supplement 4.** Pilot study for localization of [¹¹¹In]In-4497-IgG1 to subcutaneous implant-associated

*Figure 7 continued on next page*

*Figure 7 continued*

biofilm in a mouse model.

**Figure 7—animation 1.** Localization of [¹¹¹In]In-4497-IgG1 to subcutaneous implant pre-colonized with biofilm in a mouse model.

**Figure 7—animation 2.** 3D projections of [¹¹¹In]In-palivizumab injected in mice subcutaneously bearing pre-colonized and sterile catheters.

## Discussion

Identification of mAbs against *S. aureus* biofilms is a crucial starting point for the diagnosis of implant- or catheter-related infections. In this study, we show that previously identified mAbs against *S. aureus* surface structures have the capacity to bind *S. aureus* biofilm. At the start of this study, the only mAb known to react with *S. aureus* biofilm was the F598 antibody recognizing PNAG. F598 was selected to bind to planktonic *S. aureus* MN8m, which is a spontaneous PIA/PNAG-overproducing mutant of strain Mn8 (*Kelly-Quintos et al., 2006*). Because numerous studies have shown that *S. aureus* is capable of forming different biofilm matrices (PNAG-positive and PNAG-negative) (*Beenken et al., 2004*; *Moormeier et al., 2014*; *Boles et al., 2010*), we here focused on identifying antibodies recognizing different biofilm forms. Our study identified several mAbs (*Figures 5 and 6*) capable of binding both types of biofilm (4497-, CR5132-, and rF1-IgG1). This indicates that mAbs directed against WTA or the SDR protein family may be interesting candidates for targeting *S. aureus* biofilm infections. WTA comprises ~30% of the *S. aureus* bacterial surface, and therefore, it is an attractive mAb target (*Brown et al., 2013*). However, WTA glycosylation can be strain specific and *S. aureus* can adapt WTA glycosylation upon environmental cues (*Mistretta et al., 2019*). Indeed, we found that 4461-IgG1 (anti-α-GlcNAc WTA) and 4497-IgG1 (anti-β-GlcNAc WTA) recognized different *S. aureus* strains and their biofilm. Thus, mAbs targeting WTA may best be composed of a mix of mAbs recognizing both α- and β-glycosylated WTA.

Our study also shows that it is possible for antibodies to recognize both *S. aureus* biofilm and planktonic bacteria. This is crucial because during biofilm infection individual bacteria can disperse from the biofilm by secretion of various enzymes and surfactants to degrade the EPS (*Lister and Horswill, 2014*). These dispersed bacteria can then disseminate and colonize new body sites or

develop into sepsis, which is the most serious complication of biofilm-associated infections. With antibodies recognizing both biofilms and planktonic bacteria (like mAbs recognizing WTA [4461, 4497]) and SDR protein family (rF1), it should be possible to target *S. aureus* bacteria in vivo throughout the entire infection cycle (*Table 1*). We also observed that some mAbs (F598 and CR5132) bind better to biofilm than the planktonic form of *S. aureus*. Such antibodies might be useful for the development of assays to discriminate between biofilm and planktonic cultures. Importantly, none of the mAbs in our panel bound planktonic *S. aureus* but not biofilm produced by the same strain. As our data suggest that the ability to form PNAG dependent biofilm is not a hallmark of certain infections, we think it is important to identify antibodies that recognize both phenotypes. Here, we show that 4497, CR5132, rF1, and T1-2 recognize a large set of clinical isolates derived from biofilm-related infections, being PNAG-dependent and -independent. Potentially, these results can be extended to other bacterial species such as *S. epidermidis*, which is the other main cause of implant-associated infections. Three mAbs in the panel (rF1 [*Hazenbos et al., 2013*], F598 [*Kelly-Quintos et al., 2006*], CR5132 [US 2012/0141493 A1]) have been described to bind *S. epidermidis* in its planktonic state.

Altogether, our in vitro data suggested that mAbs against *S. aureus* surface antigens may be suited to detect biofilms in vivo. As a proof of principle, we tested $^{111}$In-labeled 4497-IgG1 localization to a subcutaneously implanted pre-colonized catheter in mice and found increased radiolabel around the colonized implant compared to the sterile implant within 24 hr after mAb injection, suggesting rapid localization of 4497-IgG1 to biofilm in vivo. The nonspecific localization of control-IgG1 to pre-colonized catheters at lower levels than specific IgG1 suggests that localization is a combination-specific binding to target antigens and nonspecific binding to SpA expressed by *S. aureus*. Of note, we used a pre-colonized implant model and not an infection model where biofilm is developed in vivo. In the latter model, host factors such as fibrinogen will be incorporated in the in vivo biofilm EPS (*Zapotoczna et al., 2015*; *Kwiecinski et al., 2015*; *Kwiecinski et al., 2016*; *Nishitani et al., 2015*), which is why clinical biofilm is described as very heterogenic. Therefore, it is important to further test mAb binding in different in vivo models such as PJIs and osteomyelitis models.

We consider these results as a good starting point to further evaluate the diagnostic and therapeutic purposes of these mAbs. For advanced diagnostic purposes, specific mAbs could also be coupled to gamma- or positron-emitting radionuclides and then be used to detect the presence of *S. aureus* in a biofilm in a patient or during revision surgery. Alternatively, mAbs could be used in vitro to detect the presence of biofilm on explanted implants. For therapeutic purposes, mAbs that bind to biofilm could function as a delivery vehicle to specifically direct biofilm degrading enzymes, antibiotics, photosensitizers, or alpha-/beta-emitting radionuclides to the site of infection. Alternatively, biofilm-binding mAbs could be tested for their ability to induce the activation of the immune system via the Fc-domain (*de Vor et al., 2020*). In all cases, the identification of mAbs recognizing *S. aureus* biofilm will have vast utility in the development of diagnostic and therapeutic tools for patients undergoing medical procedures.

## Materials and methods

### Key resources table

| Reagent type (species) or resource | Designation | Source or reference | Identifiers | Additional information |
|---|---|---|---|---|
| Strain, strain background (*Staphylococcus aureus*) | Wood46 | PMID:28360163 | ATCC 10832 | |
| Strain, strain background (*S. aureus*) | USA300 LAC; LAC WT (AH1263) | PMID:20418950 | AH1263 | |
| Strain, strain background (*S. aureus*) | USA300 LAC Δspa, sbi::Tn; LACΔspaΔsbi (AH4116) | This paper (*Ibberson et al., 2014*) | AH4116 | sbi::Tn in USA300 LAC Δspa (AH3052) |
| Strain, strain background (*S. aureus*) | USA300 LAC AH4802 | PMID:31723175 | AH4802 | |
| Antibody | (Goat polyclonal) anti-human IgG F(ab')₂-APC | Jackson ImmunoResearch | Cat # 109-136-088; RRID:AB_2337691 | (1:500) |

*Continued on next page*

*Continued*

| Reagent type (species) or resource | Designation | Source or reference | Identifiers | Additional information |
|---|---|---|---|---|
| Antibody | (Goat polyclonal) anti-human-kappa F(ab')₂-A647 | Southern Biotech | Cat # 2062-31; RRID:AB_2795742 | (1:500) |
| Antibody | (Humanized monoclonal) palivizumab | MedImmune | SYNAGIS | |
| Cell line | EXPI293F cell | Life Technologies | RRID:CVCL_D615 | |
| Recombinant DNA reagent | pcDNA3.4 (plasmid) | Thermo Fisher Scientific | Cat # A14697 | |
| Recombinant DNA reagent | pFUSE-CLIg-hk (plasmid) | Invivogen | Cat # pfuse2-hclk | |
| Recombinant DNA reagent | pFUSE-CHIg-hG1 (plasmid) | Invivogen | Cat # pfuse-hchg1 | |
| Recombinant DNA reagent | pFUSE-CHIg-hG3 (plasmid) | Invivogen | Cat # pfuse-hchg301 | |
| Chemical compound, drug | Bifunctional CHXA" | Macrocyclics | | |
| Chemical compound, drug | [¹¹¹In]InCl₃ | Curium Pharma | | |
| Software, algorithm | FlowJo version 10 | BD Biosciences | RRID:SCR_008520 | |
| Software, algorithm | PMOD software | PMOD Technologies | RRID:SCR_016547 | |
| Other | Syto9 stain | Invitrogen | Live/Dead BacLight Bacterial Viability Kit | (6 µM) |

## Expression and isolation of human mAbs

For human mAb expression, variable heavy (VH) and light (VL) chain sequences were cloned in home-made pcDNA3.4 expression vectors containing human heavy chain (HC) and light chain (LC) constant regions, respectively. To generate these homemade HC and LC constant region expression vectors, HC and LC constant regions from pFUSE-CHIg-hG1, pFUSE-CHIg-hG3, and pFUSE-CLIg-hk (Invivogen) were amplified by PCR and cloned separately into pcDNA3.4 (Thermo Fisher Scientific). All sequences used are shown in *Supplementary file 1*. VH and VL sequences were derived from antibodies previously described in scientific publications and patents listed in *Supplementary file 1*. Originally, all antibodies have been described as fully human, except for A120 was raised in mice by immunization with *S. aureus* LTA (EP2027155A2) and T1-2, which was raised in mice by immunization with ClfA (*Hall et al., 2003*) and later humanized to T1-2 (*Domanski et al., 2005*). CR5132 was discovered using ScFv phage libraries (US 2012/0141493 A1), and F598 (*Kelly-Quintos et al., 2006*), 4461, 4497 (*Lehar et al., 2015*), and rF1 (*Hazenbos et al., 2013*) were cloned from human B cells derived from *S. aureus*-infected patients. For each VH and VL, human codon-optimized genes with an upstream KOZAK sequence and a HAVT20 signal peptide (MACPGFLWALVISTCLEFSMA) were ordered as gBlocks (Integrated DNA Technologies) and cloned into pcDNA3.4 HC and LC constant region expression vectors using Gibson assembly (BIOKÉ). TOP10F' *Escherichia coli* were used for propagation of the generated plasmids. After sequence verification, plasmids were isolated using NucleoBond Xtra Midi plasmid DNA purification (MACHEREY-NAGEL). For recombinant antibody expression, 2 × 10⁶ cells/ml EXPI293F cells (Life Technologies) were transfected with 1 µg DNA/ml cells in a 3:2 (LC:HC) ratio and transfected using polyethylenimine HCl MAX (Polysciences). EXPI293F cells were routinely screened negative for mycoplasma contamination. After 4–5 days of expression, IgG1 antibodies were isolated from cell supernatant using a HiTrap protein A column (GE Healthcare) and IgG3 antibodies were isolated with a HiTrap Protein G High Performance column (GE Healthcare) using the Äkta Pure protein chromatography system (GE Healthcare). Antibody fractions were dialyzed overnight in PBS and filter-sterilized though 0.22 µm Spin-X filters. Antibodies were analyzed

by size-exclusion chromatography (GE Healthcare) and separated for monomeric fraction in case aggregation levels were >5%. Antibody concentration was determined by measurement of the absorbance at 280 nm and stored at –20°C.

## Bacterial strains and growth conditions

*S. aureus* strains Wood46 (ATCC 10832) (*Amend et al., 1984*; *Balachandran et al., 2017b*; *Balachandran et al., 2017a*), USA300 LAC (AH1263) (*Boles et al., 2010*), and USA300 LAC *Δspa, sbi*::Tn (AH4116) were used in this study. Strain USA300 LAC *Δspa, sbi*::Tn (AH4116) was constructed by transducing *sbi*::Tn from Nebraska Transposon Library (*Fey et al., 2013*) into USA300 LAC *Δspa* (AH3052) (*Ibberson et al., 2014*) with phage 11. Strains were grown overnight on sheep blood agar (SBA) at 37°C and were cultured overnight in Tryptic Soy Broth (TSB) before each experiment. For exponential phase planktonic cultures, overnight cultures were subcultured in fresh TSB for 2 hr. For stationary phase planktonic cultures, overnight cultures in TSB were used.

## Biofilm culture

For PNAG-negative biofilm, overnight cultures of LAC or LAC *Δspa sbi*::Tn were diluted to an $OD_{600}$ of 1 and then diluted 1:1000 in fresh TSB containing 0.5% (wt/vol) glucose and 3% (wt/vol) NaCl. 200 µL was transferred to wells in a flat-bottom 96-well plate (Corning Costar 3598, Tissue Culture treated) and incubated statically for 24 hr at 37°C. To facilitate attachment of PNAG-negative bacteria to the wells, plates were coated overnight at 4°C before inoculation. For experiments with EPS degrading enzymes, plates were coated with 20% human plasma (Sigma) in carbonate–bicarbonate buffer. For IgG1 binding assays, plates were coated with 20 µg/mL human fibronectin (Sigma) in 0.1 M carbonate–bicarbonate buffer (pH 9.6). PNAG-positive Wood46 biofilms were grown similarly, except that no coating was used and growth medium was TSB supplemented with 0.5% (wt/vol) glucose.

## Crystal violet assay

To determine the sensitivity of biofilms to DNase I, 1 mg/mL bovine DNase I (Roche) was added at the same time as inoculation and incubated during biofilm formation for 24 hr. To determine biofilm sensitivity to DspB, 30 nM DspB (MTA-Kane Biotech Inc) was added to 24 hr biofilm and incubated statically for 2 hr at 37°C. Biofilm adherence after treatment with DNase I or DspB compared to untreated controls was analyzed as follows. Wells were washed once with PBS, and adherent cells were fixed by drying plates at 60°C for 1 hr. Adherent material was stained with 0.1% crystal violet for 5 min, and excess stain was removed by washing with distilled water. Remaining dye was solubilized in 33% acetic acid, and biofilm formation was quantified by measuring the absorbance at 595 nm using a CLARIOstar plate reader (BMG LABTECH).

## Scanning electron microscopy

Biofilms were grown as described above but on 12 mm round poly-L-lysin-coated glass coverslip (Corning). Coverslips were washed 1× with PBS and fixed for 24 hr at room temperature with 2% (v/v) formaldehyde, 0.5% (v/v) glutaraldehyde, and 0.15% (w/v) Ruthenium Red in 0.1 M phosphate buffer (pH 7.4). Coverslips were then rinsed two times with phosphate buffer and post-fixed for 2 hr at 4°C with 1% osmium tetroxide and 1.5% (w/v), potassium ferricyanide (K3[Fe(CN)6]) in 0.065 M phosphate buffer (pH 7.4). Coverslips were rinsed once in distilled water followed by a stepwise dehydration with ethanol (i.e., 50%, 70%, 80%, 95%, 2 × 100%). Samples were then treated stepwise with hexamethyldisilizane (i.e., 50% HMDS/ethanol, 2 × 100% HMDS) and air-dried overnight. The next day samples were mounted on 12 mm aluminum stubs for SEM using carbon adhesive discs (Agar Scientific), and additional conductive carbon tape (Agar Scientific) was placed over part of the sample to establish a conductive path to reduce charging effects. To further improve conductivity, the surface of the samples was coated with a 6 nm layer of Au using a Quorum Q150R S sputter coater. Samples were imaged with a Scios FIB-SEM (Thermo Scientific) under high-vacuum conditions at an acceleration voltage of 20 kV and a current of 0.40 nA.

## Antibody binding to planktonic cultures

To determine mAb binding capacity, planktonic bacterial cultures were suspended and washed in PBS containing 0.1% BSA (Serva) and mixed with a concentration range of IgG1-mAbs in a round-bottom

96-well plate in PBS-BSA. Each well contained $2.5 \times 10^6$ bacteria in a total volume of 55 µL. Samples were incubated for 30 min at 4°C, shaking (~700 rpm), and washed once with PBS-BSA. Samples were further incubated for another 30 min at 4°C, shaking (~700 rpm), with APC-conjugated polyclonal goat-anti-human IgG F(ab')$_2$ antibody (Jackson ImmunoResearch, 1:500). After washing, samples were fixed for 30 min with cold 1% paraformaldehyde. APC fluorescence per bacterium was measured on a flow cytometer (FACSVerse, BD). Control bacteria were used to set proper FSC and SSC gate definitions to exclude debris and aggregated bacteria. Data were analyzed with FlowJo (version 10).

### Antibody binding to biofilm cultures

To determine mAb binding capacity to biofilm, wells containing 24 hr biofilm were blocked for 30 min with 4% BSA in PBS. After washing with PBS, wells were incubated with a concentration range of IgG1-mAbs, or Fab fragments when indicated, in PBS-BSA (1%) for 1 hr at 4°C, statically. After washing two times with PBS, samples were further statically incubated for 1 hr at 4°C with APC-conjugated polyclonal goat-anti-human IgG F(ab')$_2$ antibody (Jackson ImmunoResearch, 1:500). Fab fragments were detected with Alexa Fluor 647-conjugated goat-anti-human-kappa F(ab')$_2$ antibody (Southern Biotech, 1:500). After washing, fluorescence per well was measured using a CLARIOstar plate reader (BMG LABTECH).

### Peptidoglycan and LTA ELISA

Peptidoglycan from Wood46 was isolated as described in *Timmerman et al., 1993*, and purified LTA was a kind gift from Sonja von Aulock and Siegfried Morath (University of Konstanz). We coated Maxi-sorb plates (Nunc) overnight at 4°C with 1 µg/mL peptidoglycan or LTA. The plates were washed three times with PBS 0.05% Tween, blocked with PBS 4% BSA, and incubated 1 hr with a concentration range of CR5132-IgG1, A120-IgG1 (directed against LTA), or control IgG1. The plates were washed and incubated 1 hr with 1:6000 goat-fab'2-anti-human-kappa-HRP (Southern Biotech). Finally, the plates were washed and developed using 3,3',5,5'-tetramethylbenzidine (Thermo Fisher). The reaction was stopped by addition of 1 N $H_2SO_4$. Absorption at 450 nm was measured using a CLARIOstar plate reader (BMG LABTECH).

### IgG1 binding to WTA glycosylated beads

Synthetic WTA (a kind gift of Jeroen Codee, Leiden University) was immobilized on magnetic beads as in *van Dalen et al., 2019*. Shortly, biotinylated RboP hexamers were enzymatically glycosylated by recombinant TarM, TarS, or TarP with UDP-GlcNAc (Merck) as substrate. After 2 hr incubation at room temperature, $5 \times 10^7$ pre-washed Dynabeads M280 Streptavidin (Thermo Fisher) were added and incubated for 15 min at room temperature. The coated beads were washed three times in PBS using a plate magnet, resuspended in PBS 0.1% BSA, and stored at 4°C. To determine CR5132 binding capacity, beads were suspended and washed in PBS/0.05% Tween/0.1% BSA and mixed with a concentration range of CR5132-IgG1 or control IgG1 in a round-bottom 96-well plate in PBS/Tween/BSA. Each well contained $10^5$ beads. Samples were incubated for 30 min at 4°C, shaking (~700 rpm), and washed once with PBS/Tween/BSA. Samples were further incubated for another 30 min at 4°C, shaking (~700 rpm), with APC-conjugated polyclonal goat-anti-human IgG F(ab')$_2$ antibody (Jackson ImmunoResearch, 1:500). After washing, APC fluorescence per bead was measured on a flow cytometer (FACSVerse, BD).

### Antibody binding in the presence of human pooled IgG

MAb binding in the presence of human pooled IgG was assessed with mAbs that were directly fluorescently labeled. Briefly, mAbs were labeled with AF647 NHS ester (Thermo Fisher Scientific) by following the manufacturer's protocol. Labeled mAbs were buffer exchanged into PBS using desalting Zeba columns (Thermo Fisher Scientific), checked for degree of labeling (ranging from 2.9 to 4.5), and stored at 4°C. To isolate human pooled IgG, blood was drawn from 22 healthy volunteers and allowed to clot for 15 min at room temperature. After centrifugation for 10 min at $3220 \times g$ at 4°C, serum was collected, pooled, and subsequently stored at –80°C. IgG was isolated from pooled serum as described above. Biofilm cultures were prepared, washed, and incubated as described above. Samples were incubated with 10 µg/mL AF647-conjugated IgG1 mAbs in buffer or buffer containing

250 µg/mL pooled IgG. AF647 fluorescence per well was measured using a CLARIOstar plate reader (BMG LABTECH).

## Confocal microscopy of static biofilm

Wood46 and LAC Δspa sbi::Tn biofilm were grown in glass-bottom cellVIEW slides (Greiner Bio-One [543079]) similarly as described above. cellVIEW slides were placed in a humid chamber during incubation to prevent evaporation of growth medium. After 24 hr, wells were gently washed with PBS and fixed for 30 min with cold 1% paraformaldehyde, followed by blocking with 4% BSA in PBS. After washing with PBS, wells were incubated with 66 nM IgG1-mAbs in PBS-BSA (1%) for 1 hr at 4°C, statically. After washing two times with PBS, samples were further statically incubated for 1 hr at 4°C with Alexa Fluor 647-conjugated goat-anti-human-kappa F(ab')$_2$ antibody (Southern Biotech, 1:300) and 6 µM Syto9 (Live/Dead BacLight Bacterial Viability Kit; Invitrogen). Z-stacks at three random locations per sample were collected at 0.42 µm intervals using a Leica SP5 confocal microscope with a HCX PL APO CS 63×/1.40–0.60 OIL objective (Leica Microsystems). Syto9 fluorescence was detected by excitation at 488 nm, and emission was collected between 495 nm and 570 nm. Alexa Fluor 647 fluorescence was detected by excitation at 633 nm, and emission was collected between 645 and 720 nm. Image acquisition and processing was performed using Leica LAS AF imaging software (Leica Microsystems).

## Subcutaneous implantation of pre-colonized catheters in mice

To determine in vivo mAb localization to implant-associated biofilm, we subcutaneously implanted pre-colonized catheters in mice, as described in *Kadurugamuwa et al., 2003*. Balb/cAnNCrl male mice weighing >20 g obtained from Charles River Laboratories were housed in our Laboratory Animal Facility. 1 hr before surgery, all mice were given 5 mg/kg carprofen. Anesthesia was induced with 5% isoflurane and maintained with 2% isoflurane. Their backs were shaved and the skin was disinfected with 70% ethanol. A 5 mm skin incision was made using scissors after which a 14 gauge piercing needle was carefully inserted subcutaneously at a distance of approximately 1–2 cm. A 5 mm segment of a 7 French polyurethane catheter (Access Technologies) was inserted into the piercing needle and correctly positioned using a k-wire. The incision was closed using one or two sutures, and the skin was disinfected with 70% ethanol. Mice received one s.c. catheter in each flank. One catheter served as a sterile control, whereas the other was pre-colonized for 48 hr with an inoculum of ~10$^7$ CFU *S. aureus* LAC AH4802. Strain AH4802 is identical to AH4807 as reported in *Miller et al., 2019*. The implantation of sterile and pre-colonized catheters in the left or the right flank was randomized. Before inoculation, the implants were sterilized with 70% ethanol and air dried. The inoculated implants were incubated at 37°C for 48 hr under agitation (200–300 RPM). New growth medium was added at 24 hr to maintain optimal growing conditions. Implants were washed three times with PBS to remove nonadherent bacteria and stored in PBS until implantation or used for determination of viable CFU counts. To this end, implants were placed in PBS and sonicated for 10 min in a Branson M2800E Ultrasonic Waterbath (Branson Ultrasonic Corporation). After sonication, total viable bacterial counts per implant were determined by serial dilution and plating.

## Radionuclides and radiolabeling of antibodies

4497-IgG1 (anti-β-GlcNAc WTA) and control IgG1 antibody palivizumab (MedImmune) were labeled with indium-111 ($^{111}$In) using the bifunctional chelator CHXA″ as described previously by *Allen et al., 2018*. In short, antibodies were buffer exchanged into conjugation buffer and incubated at 37°C for 1.5 hr with a fivefold molar excess of bifunctional CHXA″ (Macrocyclics, prepared less than 24 hr before use). The mAb-CHXA″ conjugate was then exchanged into 0.15 M ammonium acetate buffer to remove unbound CHXA″ and subsequently incubated with approximately 150 kBq $^{111}$In (purchased as [$^{111}$In]InCl$_3$ from Curium Pharma) per µg mAb. The reaction mixture was incubated for 60 min at 37°C after which free $^{111}$In$^{3+}$ was quenched by the addition of 0.05 M EDTA. Quality control was done by instant thin layer chromatography (iTLC) and confirmed radiolabeling at least 95% radiochemical purity of the antibodies.

## USPECT-CT and CFU count

G*power 3.1.9.2 software was used to estimate group sizes for mouse experiments, aiming for a power of 0.95. A minimum of four mice per group was calculated based on the expected difference between 4497-IgG1 localization to sterile implants versus pre-colonized implants and experimental variation obtained in a pilot study. In the event that mAbs were incorrectly injected into the tail vain, mice were excluded from the analyses. Incorrect injection was determined by visual inspection during injection and with SPECT/CT scan, showing radioactivity in the tail tissue instead of the bloodstream.

Two days after subcutaneous implantation of catheters, 50 µg radiolabeled antibody (7.5 MBq) was injected into the tail vein. Four mice were injected with $[^{111}In]$In-4497-IgG1 and four mice were injected with $[^{111}In]$In-palivizumab. At 24, 72, and 120 hr post injection, multimodality SPECT/CT imaging of mice was performed with a VECTor[6] CT scanner (MILabs, The Netherlands) using a MILabs HE-UHR-M mouse collimator with 162 pinholes (diameter, 0.75 mm) (*Goorden et al., 2013*). At 24 hr, a 30 min total-body SPECT-CT scan was conducted under anesthesia. Scanning duration at 72 and 120 hr was corrected for the decay of $^{111}$In. Immediately after the last scan, mice were sacrificed by cervical dislocation while under anesthetics. The carcasses were stored at −20°C until radiation exposure levels were safe for further processing. Implants were aseptically removed, placed in PBS, and sonicated for 10 min in a Branson M2800E Ultrasonic Waterbath (Branson Ultrasonic Corporation). After sonication, total viable bacterial counts per implant were determined by serial dilution and plating.

## Image visualization and SPECT/CT data analyses

The analyzing investigator was blinded for the injection of $[^{111}In]$–4497-IgG1 or $[^{111}In]$-palivizumab. Image processing and volume of interest analysis of the total-body SPECT scans were done using PMOD software (PMOD Technologies). SPECT image reconstruction was performed using Similarity Regulated OSEM (*Vaissier et al., 2016*), using 6 iterations and 128 subsets, and the total-body SPECT volumes were smoothed using a 3D Gaussian filter of 1.5 mm. To quantify the accumulation of $^{111}$In around the catheters, regions of interest (ROIs) were delineated on SPECT/CT fusion scans as in *Branderhorst et al., 2014*. 2D ROIs were manually drawn around the catheters and the full body on consecutive transversal slices that were reconstructed into a 3D volume of interest. Delineating the ROIs was done using an iso-contouring method with a threshold of 0.11. For each ROI, the reconstructed voxel intensity sums (total counts) were related to calibrator dose measurements (kBq). Accumulation of $^{111}$In was defined as a percentage of total body activity, calculated as (total activity in the implant ROI/total activity in the body ROI) * 100. Reconstructed 3D body scans were visualized as maximum intensity projections, and the SPECT scale was adjusted by cutting 10% of the lower signal intensity to make the high-intensity regions readily visible.

## Statistical testing

Statistical analyses were performed in GraphPad Prism 8. The tests and n-values used to calculate p-values are indicated in the figure legends. Unless stated otherwise, graphs comprised at least three biological replicates (independent experiments). When indicated, experiments were performed with technical replicates (duplicate/triplicate).

## Acknowledgements

The authors greatly thank Reindert Nijland (Department of Animal Sciences, Wageningen University and Research, Wageningen, The Netherlands) and Fernanda Paganelli (Department of Medical Microbiology, UMC Utrecht) for assistance with biofilm work; Miquel Ekkelenkamp and Sebastian van Marm (Department of Medical Microbiology, UMC Utrecht) for providing *S. aureus* clinical isolates; Astrid Hendriks (Department of Medical Microbiology, UMC Utrecht), Nina van Sorge (Department of Medical Microbiology and Infection Prevention, Amsterdam UMC, The Netherlands), and Jeroen Codee (Leiden Institute of Chemistry, Leiden University, The Netherlands) for providing WTA glycosylated beads; Frank Beurskens (Genmab BV, Utrecht, The Netherlands) for help with selection of mAbs; Sonja von Aulock and Siegfried Morath (Department of Biochemical Pharmacology, University of Konstanz, Konstanz, Germany) for providing LTA preparates. LdV and BvD were supported by a grant from Health~Holland (LSHM17026 to JvS and HW). FJB and RMR were supported by the research grant QUARAT: Quantitative Universal Radiotracer Tomography (TTW16885, Dutch Research

Council (NWO)). ARH was supported by a merit award (BX002711) from the U.S. Department of Veteran Affairs and grant AI083211 from the National Institutes of Health.

## Additional information

### Competing interests
Ruud M Ramakers: Ruud M. Ramakers has stock appreciation rights at MILabs B.V. Freek J Beekman: Freek J. Beekman is shareholder at MILabs B.V. The other authors declare that no competing interests exist.

### Funding

| Funder | Grant reference number | Author |
| --- | --- | --- |
| Health~Holland | LSHM17026 | Lisanne de Vor<br>Bruce van Dijk<br>Jos van Strijp<br>Harrie Weinans |
| National Institutes of Health | AI083211 | Alexander R Horswill |
| QUARAT: Quantitative Universal Radiotracer Tomography | TTW16885 | Freek J Beekman<br>Ruud M Ramakers |
| U.S. Department of Veteran Affairs | BX002711 | Alexander R Horswill |

The funders had no role in study design, data collection and interpretation, or the decision to submit the work for publication.

### Author contributions
Lisanne de Vor, Bruce van Dijk, Conceptualization, Formal analysis, Investigation, Methodology, Visualization, Writing – original draft; Kok van Kessel, Conceptualization, Investigation, Methodology, Supervision, Writing – original draft; Jeffrey S Kavanaugh, Edwin C Boel, Ad C Fluit, Jakub M Kwiecinski, Gerard C Krijger, Ruud M Ramakers, Freek J Beekman, Ekaterina Dadachova, Marnix GEH Lam, H Charles Vogely, Bart CH van der Wal, Alexander R Horswill, Conceptualization, Writing – review and editing; Carla de Haas, Piet C Aerts, Marco C Viveen, Conceptualization, Investigation, Methodology, Writing – review and editing; Jos AG van Strijp, Conceptualization, Funding acquisition, Writing – review and editing; Harrie Weinans, Conceptualization, Funding acquisition, Supervision, Writing – original draft; Suzan HM Rooijakkers, Conceptualization, Supervision, Writing – original draft, Writing – review and editing

### Author ORCIDs
Lisanne de Vor (iD) http://orcid.org/0000-0002-2289-3378
Edwin C Boel (iD) http://orcid.org/0000-0002-8506-0357
Jakub M Kwiecinski (iD) http://orcid.org/0000-0001-9472-2896
Ekaterina Dadachova (iD) http://orcid.org/0000-0001-7300-6479
Suzan HM Rooijakkers (iD) http://orcid.org/0000-0003-4102-0377

### Ethics
All animal procedures were approved by the Utrecht University animal ethics committee and were performed in accordance with international guidelines on handling laboratory animals (Animal Use Permit #AVD1150020174465, approved 1 March 2018). To obtain human serum, blood was isolated from healthy donors according to a study protocol approved by the Medical Ethics Committee of the University Medical Center Utrecht was obtained (METC protocol 07-125/C approved on March 1, 2010).

### Decision letter and Author response
Decision letter https://doi.org/10.7554/eLife.67301.sa1

Author response https://doi.org/10.7554/eLife.67301.sa2

## Additional files

### Supplementary files
- Supplementary file 1. Protein sequences used for human monoclonal antibody production.
- Supplementary file 2. Exact p-values and statistical analysis performed per figure.
- Transparent reporting form

### Data availability
All data generated or analysed during this study are included in the manuscript and supporting files. Supporting data has been uploaded to Dryad.

The following dataset was generated:

| Author(s) | Year | Dataset title | Dataset URL | Database and Identifier |
|---|---|---|---|---|
| de Vor L | 2021 | Source data for: Human monoclonal antibodies against Staphylococcus aureus surface antigens recognize in vitro biofilm and in vivo implant infections | https://doi.org/ 10.5061/dryad. vmcvdncs2 | Dryad Digital Repository, 10.5061/dryad.vmcvdncs2 |

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
