## [Editor Report]

This article deals with the development of novel antibodies that are able to attach on both the planktonic and replicating phase of biofilm formed by *S. aureus*. The authors present evidence for the attachment ability of these antibodies using both in vitro experiments and animal experiments.

---

## [Decision Letter]

**Decision letter after peer review:**

Thank you for submitting your article "Human monoclonal antibodies against *Staphylococcus aureus* surface antigens recognize biofilm and implant infections" for consideration by *eLife*. Your article has been reviewed by 3 peer reviewers, including Evangelos J Giamarellos-Bourboulis as the Reviewing Editor and Reviewer #1, and the evaluation has been overseen by Gisela Storz as the Senior Editor. The following individual involved in review of your submission has agreed to reveal their identity: Andreas Peschel (Reviewer #3).

The authors take advantage of published sequences of 6 *S. aureus*-specific antibodies. 5 of these antibodies target surface determinants of *S. aureus*. The authors uncover the true target for the 6th antibody erroneously assumed to be directed against LTA. The authors conduct a rigorous analysis of these antibodies. All CDRs are cloned using the same hIgG1 framework and all recombinant antibodies are produced using the same cell system. The authors include appropriate isotype controls.

The study is complicated by the fact that not all strains of *S. aureus* produce biofilms through the same mechanisms (at least in vitro). Here the authors chose two "representative" strains: Wood 46, a PNAG-dependent biofilm former and LAC, a PNAG-independent biofilm former. More critically, the study is restricted to strains that produce less or no IgG binding proteins, SpA and Sbi. This diminishes the impact of the study. SpA and Sbi are displayed in the bacterial envelope and capture IgG1 non-specifically through the Fc domain.

By limiting the analysis to spa/sbi deficient strains, the authors find that antibodies specific for β-GlcNac-WTA and SDR proteins can identify both PNAG-positive and PNAG-negative biofilms in vitro. Anti-SDR antibodies are also able to bind planktonic bacteria. The authors validate their findings by using β-GlcNac-WTA antibodies to detect biofilms in a mouse model. They present some very nice images and videos. In principle, the authors achieved their goals i.e. a proof-of-concept study for antibodies that can "mark" biofilms of *S. aureus*.

Essential revisions:

• The authors need to provide evidence about the power of their animal study. Using one group of two mice introduces severe limitations.

• There is no evidence that mice developed osteomyelitis. Histology is missing. The authors need also to explain why the classical osteomyelitis model of Mader et al. was not studied in their setting.

• The authors need to omit from their manuscript the use of their antibodies for treatment; they do not provide any evidence for therapy.

• l.95-96: here the authors state that PNAG-dependent (+) biofilms are phenotypically different from PNAG-independent (-) biofilms and cite 6 references. The authors do not explain whether these phenotypic differences correlate with specific clinical biofilms. In other words, are PNAG+ biofilms a hallmark of infective endocarditis, catheter-associated infections, or arthritis? Is there any correlation at all? At the very least, the authors should test their hypothesis in the animal model. For instance, how would antibody F598 behave in the experiment shown in Figure 6? Would F598 selectively label the Woods46 biofilm but not LAC biofilms?

• Presumably, the reduced surface display of SpA in strain Wood46 results from decreased expression of the srtA gene (i.e. failure to anchor SpA to peptidoglycan). Nonetheless, Wood46 continues to secrete SpA (Balachandran et al., Plos one 2017). In Figure 2a, hIgG1 control and secondary goat antibodies do not bind to planktonic bacteria. However, control hIgG1 seems to bind Wood46 biofilms quite well (Figure 2B, Figure 3B, Figure 4B, Figure 5A). How do the authors explain the data? Is secreted SpA diffusing away from planktonic cells but retained in biofilms?

• The authors should use wild type LAC (or a wild type strain Sbi+/Spa+) to evaluate anti-β-GlcNac-WTA and anti-SDR antibodies under physiological conditions.

• In Figure S1a, it looks as if all strains can attach to uncoated polystyrene microtiter plates however the text indicates that PNAG(+) strains preferentially adhere to such plates (l. 121-123). Is there a significant difference between Wood46 and USA300 LAC in Figure S1a?

• Figure 1 – S1a: in this study, the authors include either PNAG-dependent or DNA-dependent biofilm strains. Is this the sum of all biofilms encountered in the clinic? Does this imply that antibodies against β-GlcNac-WTA and SDR proteins should be able to mark and identify all biofilms (PNAG+/- and DNA+/-)?

• The use of signs "+" and "-" in Table 1 isn't particularly informative. Could the authors quantify or better describe weak and strong binding activities?

• Figure 6A: could the authors provide a better assessment for the limit of detection? How large should the biofilm be for visualization shown on Figure 6? Overall the images and videos are aesthetically pleasing, yet circling the area encompassing the biofilm and site of injection is a bit obvious. Were the images analyzed by a "blind" investigator and did the blind investigator reach the same conclusion as the authors?

• Figure 6A: the authors note that the overall antibody intensity decreases quite dramatically over 3 days. Does radiolabeling increase the turn-over of human IgG1 in mice? Or is the biofilm/infection responsible for the increased turn-over? *S. aureus* secretes several proteases that can degrade antibodies. This could be easily tested by using distinct animals for sterile and pre-colonized implants, although it is understood if adding additional animals to this particular study might be challenging.

• Line 113-116: Not clear if the statement refers to the control mAbs only or also to the tested mAbs.

• Line 125: Please rephrase sentence, as it can currently be read as either "SpA complicates the detection…" or "low surface expression of SpA complicates the detection…"

• Line 130-131: Please add a reference here.

• 134-135: Does this mean that (1) there is no DNA present in this biofilm, (2) the DNA is not digested by DNaseI due to shielding by other structural components (like PNAG) or (3) that digestion of DNA by DNaseI does not disturb the biofilm as other structural components (like PNAG) are present?

• 148: Growth condition affects regulation of the enzymes, and thereby their presence. I would suggest rephrasing the sentence to include both (1) presence of the genes encoding the responsible glycosyltransferase enzymes, as well as (2) expression of these genes based on environmental conditions (such as growth phase). Cite e.g. Li et al. 2015 (DOI: 10.1038/srep17219) regarding point 1.

• 153: This is related to the used growth phase of the bacterial cells (24h), which features high tarM expression. I would suggest also performing this experiment using exponential phase cultures, similar to how this is also performed for the ClfA experiments later on, in order to avoid any confusion regarding presence of the epitope on this strain in the literature.

• 161-164: See above. This is only true when comparing with stationary phase planktonic growth in broth.

• 178-179: The authors may consider using a PNAG-pos strain with both WTA modifications and an isogenic ica knockout to substantiate this claim.

• 192-194: How was this objectively measured? By comparison of the centers of mass of the fluorescent signals? If so, it would be helpful to provide an appropriate statistical method to substantiate these claims.

• 198.201: This experiment needs some more explanation. For instance, an additional mAb A120 was used in Figure S5c, which should be described. If it is indeed WTA b-GlcNAc that is bound by CR5132, please also show deposition on exponential-phase planktonic LAC.

• 217- 219: The authors to not provide compelling evidence that the abundance of the ClfA is different between PNAG-pos and PNAG-neg biofilms. They should tone down their statement or, if they want to make such claims, should show this using a different method, e.g. western blot or on the transcriptional level by qPCR.

• 233-235: What is this ratio based on? Please describe. Not taking into account kinetics, one would need to administer at least 10 g/l * 5 l / 25 = 2 gram of monoclonal antibody to reach a similar ratio in the human circulation. Is that realistic? If not, what would be a more appropriate ratio to test in vitro?

• 274: How do the authors conclude this, as only one other body site seems to be checked (namely, the sterile control implant)?

• 275-276: Which data are the authors correlating to make this statement? Please test appropriately.

• 283-285: What about S. epidermidis? Isn't that a similar if not bigger problem regarding these sorts of infection?

• References: Text duplication in all references containing "*Staphylococcus aureus*": in all cases written as "*Staphylococcus aureusStaphyloccus aureus*"

• Figures: Please plot two-sided error bars instead of the top one only. In addition, several y-axes are too short and clip error-bars (e.g. 3b), please double-check.

• 755-756: What does control refer to? No Ab or one of the isotype control antibodies defined in Figure 1A?

*Reviewer #1:*

This manuscript deals with the development of novel antibodies that are able to attach on both the planktonic and replicating phase of biofilm formed by *S. aureus*. The authors present evidence for the attachment ability of these antibodies using both in vitro experiments and animal experiments.

The strengths of the manuscript are (a) the novel approach for an unmet need; (b) the development of antibodies that are targeting both sessile and planktonic forms; and (c) the study in an animal model. The main weak points are: (a) the lack of experimentation with coagulase-negative staphylococci that are the main effectors of biomembranes attaching on matrixes; and (b) the lack of proof that the animal model is a real model of osteomyelitis.

*Reviewer #2:*

The authors take advantage of published sequences of 6 *S. aureus*-specific antibodies. 5 of these antibodies target surface determinants of *S. aureus*. The authors uncover the true target for the 6th antibody erroneously assumed to be directed against LTA. The authors conduct a rigorous analysis of these antibodies. All CDRs are cloned using the same hIgG1 framework and all recombinant antibodies are produced using the same cell system. The authors include appropriate isotype controls.

The study is complicated by the fact that not all strains of *S. aureus* produce biofilms through the same mechanisms (at least in vitro). Here the authors chose two "representative" strains: Wood 46, a PNAG-dependent biofilm former and LAC, a PNAG-independent biofilm former. More critically, the study is restricted to strains that produce less or no IgG binding proteins, SpA and Sbi. This diminishes the impact of the study. SpA and Sbi are displayed in the bacterial envelope and capture IgG1 non-specifically through the Fc domain.

By limiting the analysis to spa/sbi deficient strains, the authors find that antibodies specific for β-GlcNac-WTA and SDR proteins can identify both PNAG-positive and PNAG-negative biofilms in vitro. Anti-SDR antibodies are also able to bind planktonic bacteria. The authors validate their findings by using β-GlcNac-WTA antibodies to detect biofilms in a mouse model. They present some very nice images and videos. In principle, the authors achieved their goals i.e. a proof-of-concept study for antibodies that can "mark" biofilms of *S. aureus*.

*Reviewer #3:*

This manuscript characterizes monoclonal antibodies directed against a variety of *S. aureus* surface molecules in terms of their capacity to detect planktonic vs. biofilm-forming *S. aureus* cells. The approach leads to a number of important results: it reveals which of the targeted surface structure and modification is expressed in in different *S. aureus* lifestyles and regions of a biofilm and it recommends the use of certain mABs for in-vivo detection of *S. aureus* biofilms.

The study also raises a number of questions that could be more extensively addressed in the discussion. For instance, the mAb binding studies reveal that α and β GlcNAc-modification probably occurs differently in planktonic vs. sessile *S. aureus* lifestyle and in different areas of a biofilm. Can these findings be explained by differential expression of tarS/P and tarM referring to previous genome-wide transcriptomics studies?

It is also remarkable that mAbs 4497 and CR5132 detect the same WTA epitope but appear to differ in their biofilm-staining capacities. Any potential explanation?

---

## [Author Response]

Essential revisions:• The authors need to provide evidence about the power of their animal study. Using one group of two mice introduces severe limitations.

We agree with the referee that the group size in our animal study was too small, therefore we have repeated the animal study to introduce more mice. The animal study was powered on the expected difference in localization of 4497-IgG1 to a sterile versus a pre-colonized implant in the same mouse, which functioned as an internal control. We repeated the experiment by injecting 5 mice with 4497-IgG1 (Figure 7, Figure 7—figure supplement 2) and 2 with ctrl-IgG1 (Figure 7—figure supplement 3). In the revised manuscript we now meet the calculated sample size of 4 mice per group (Figure 7). This information about power calculation was added to the revised manuscript in 525. We also rewrote the paragraph about this figure (line 297-314) and removed old Figure 6c as the absolute data is now more convincing and sufficient to substantiate our claims.

• There is no evidence that mice developed osteomyelitis. Histology is missing. The authors need also to explain why the classical osteomyelitis model of Mader et al. was not studied in their setting.

We understand that we did not clearly explain in the manuscript that we are not using an osteomyelitis infection model in our study. We also realize that indeed we are not using an infection model here because we are implanting implants pre-colonized with biofilm. Therefore, we more clearly explained the model and removed terminology related to ”infection” at several places in the manuscript, including the title, as indicated by track changes. Moreover, we added a line 401 in the discussion, emphasizing the importance of testing the results in a prosthetic joint infection or osteomyelitis model.

• The authors need to omit from their manuscript the use of their antibodies for treatment; they do not provide any evidence for therapy.

The referee is right that our results do not directly result in new antibody therapies. Therefore, we removed some references to the use of antibodies for treatment. However, in particular cases (introduction, discussion) we think it is relevant to refer to the potential use of mAbs as a platform in the development of new treatment. For instance, monoclonal antibodies (mAb) could be exploited as vehicles to specifically bring anti-biofilm agents (such as radionuclides, antibiotics) to the site of infection. Our results are an important first step for such developments.

• l.95-96: here the authors state that PNAG-dependent (+) biofilms are phenotypically different from PNAG-independent (-) biofilms and cite 6 references. The authors do not explain whether these phenotypic differences correlate with specific clinical biofilms. In other words, are PNAG+ biofilms a hallmark of infective endocarditis, catheter-associated infections, or arthritis? Is there any correlation at all? At the very least, the authors should test their hypothesis in the animal model. For instance, how would antibody F598 behave in the experiment shown in Figure 6? Would F598 selectively label the Woods46 biofilm but not LAC biofilms?

We agree with the referee that this is an important point, and therefore performed additional experiments to study whether PNAG production is associated with certain clinical biofilms. In literature, no correlation between *S. aureus* biofilm phenotypes and the source of clinical biofilm infections has been described. Therefore, we collected *S. aureus* isolates from endocarditis, prosthetic joint infections and catheter tip infections. Next, we used the anti-PNAG antibody (F598-IgG3) to determine if these isolates produced PNAG-positive or PNAG-negative biofilm. We could detect F598-IgG3 binding to 1/4 endocarditis isolates, 6/25 catheter tip isolates and 5/16 PJI isolates**.**

Overall, these experiments indicate that formation of PNAG positive or negative biofilm is not a hallmark of one specific source of biofilm related infections and that approximately one third of isolates form PNAG-positive biofilm in vitro. In the revised manuscript, these data are included as Figure 6 and we added a new paragraph to the Results section entitled “The majority of mAbs recognize PNAG-positive and PNAG-negative formed by clinical isolates from biofilm associated infections”(line 268 – 297).

The above data underline that it is important to identify mAbs that recognize both PNAG-positive and PNAG-negative biofilm. For this reason, we focused our in vivo experiments on a mAb that can recognize both phenotypes and not on distinguishing between the two phenotypically different biofilm. We apologize that this was not clear and now better explain that the aim of our study is to identify mAbs that recognize both types of biofilm (line 100 and 288).

• Presumably, the reduced surface display of SpA in strain Wood46 results from decreased expression of the srtA gene (i.e. failure to anchor SpA to peptidoglycan). Nonetheless, Wood46 continues to secrete SpA (Balachandran et al., Plos one 2017). In Figure 2a, hIgG1 control and secondary goat antibodies do not bind to planktonic bacteria. However, control hIgG1 seems to bind Wood46 biofilms quite well (Figure 2B, Figure 3B, Figure 4B, Figure 5A). How do the authors explain the data? Is secreted SpA diffusing away from planktonic cells but retained in biofilms?

We thank the referee for raising this interesting point. The revised manuscript contains new experimental data explaining the binding of control hIgG1 to Wood46 biofilms. To test the reviewers’ hypothesis, we performed a binding assay on Wood46 planktonic bacteria vs. biofilm with control IgG1, which is able to bind SpA via the Fc-domain, control IgG3, which is unable to bind to SpA via the Fc-domain and IgG3 mAb binding SpA via the Fab-domain but not the Fc-domain. None of the mAbs bound to planktonic bacteria (Figure 2—figure supplement 2b), indicating that SpA is removed during the washing steps in the assay. In contrast, control IgG1 showed increased binding to Wood46 biofilm compared to control IgG3 and aSpA-IgG3 showed increased binding compared to control IgG1 (Figure 2—figure supplement 2a), indicating that SpA is incorporated in the biofilm and cannot be washed away in the assay. In the revised manuscript, we added one new paragraph to the Results section (line 165-173).

• The authors should use wild type LAC (or a wild type strain Sbi+/Spa+) to evaluate anti-β-GlcNac-WTA and anti-SDR antibodies under physiological conditions.

We agree that this is a valuable addition so as suggested by the referee, we have now included mAb binding data to wild type LAC (AH1263). To exclude nonspecific binding of the IgG-Fc domain by SpA, we produced all antibodies as IgG3 subclass. Binding of IgG3 mAbs to planktonic LAC WT (Figure 5—figure supplement 4) was similar to binding of IgG1 mAbs to LAC∆spa∆sbi (Figure 5—figure supplement 1b), except for a relatively high binding of 4497-IgG3 to LAC WT compared to 4497-IgG1 binding to LAC∆spa∆sbi. This could indicate that knocking out SpA and Sbi influenced the production of β-WTA by LAC WT. Binding of IgG3 mAbs to LAC WT biofilm (Figure 5—figure supplement 4b) was comparable to binding of IgG1 mAbs to LAC*∆spa∆sbi* biofilm (Figure 5b). In the revised manuscript we added line 265-272.

• In Figure S1a, it looks as if all strains can attach to uncoated polystyrene microtiter plates however the text indicates that PNAG(+) strains preferentially adhere to such plates (l. 121-123). Is there a significant difference between Wood46 and USA300 LAC in Figure S1a?

The initial screen in old figure S1A was based on literature describing PNAG facilitating adherence to plastic via electrostatic interactions (Arciola et al., 2015; Formosa-Dague et al., 2016). However, in the clinical isolate screen we performed for point 4, we have compared biofilm formation for each isolate on uncoated versus coated plates. From this experiment it became clear that also isolates without PNAG can grow robust biofilm on uncoated plates and isolates with PNAG can form robust biofilm on coated plates. We therefore removed this argument and figure S1A from the manuscript.

• Figure 1 – S1a: in this study, the authors include either PNAG-dependent or DNA-dependent biofilm strains. Is this the sum of all biofilms encountered in the clinic? Does this imply that antibodies against β-GlcNac-WTA and SDR proteins should be able to mark and identify all biofilms (PNAG+/- and DNA+/-)?

In literature, a clear distinction is made between PNAG-positive and PNAG- negative *S. aureus* biofilm (Fitzpatrick et al., 2005; McCarthy et al., 2015; Mlynek et al., 2020; O’Neill et al., 2007; Rohde et al., 2007; Sugimoto et al., 2018). Besides PNAG, the in vitro biofilm can be composed of bacterial proteins and bacterial DNA. The relative contribution of each factor differs per *S. aureus* strain. Furthermore, in an in vivo biofilm human factors will be incorporated, which is why biofilm encountered in the clinic is highly heterogenic (Zapotoczna et al., 2015).

To answer the referees second question, from the screened clinical isolates (that were used in Figure 6ab), we selected 6 PNAG-positive and 6 PNAG-negative clinical isolates. To this selection of isolates, we tested binding of the mAb panel and we found that all mAbs, except 4461 and F598 bind to in vitro formed biofilm by PNAG-positive and PNAG-negative clinical isolates. These data have been included in Figure 6c and a new paragraph in the Results section entitled: “The majority of mAbs recognize PNAG-positive and PNAG-negative formed by clinical isolates from biofilm associated infections”.

• The use of signs "+" and "-" in Table 1 isn't particularly informative. Could the authors quantify or better describe weak and strong binding activities?

As requested by the referee, we described weak and strong binding activities as indicated with “+”, weak binding (p > 0.05 – p < 0.99) is indicated with “+/-“ and no significant binding (p > 0.99) is indicated with “–“ (line 1096).

• Figure 6A: could the authors provide a better assessment for the limit of detection? How large should the biofilm be for visualization shown on Figure 6? Overall the images and videos are aesthetically pleasing, yet circling the area encompassing the biofilm and site of injection is a bit obvious. Were the images analyzed by a "blind" investigator and did the blind investigator reach the same conclusion as the authors?

We did not assess the limit of detection. We measured the biofilm size (CFU counts) on the catheters before implantation and at the end point of the study (120h after implantation), but not at each SPECT/CT scanning time point. Therefore, the best assessment for the limit of detection we can provide is that the biofilm size was 4.5 x 10^7^ CFU/implant before implantation and 1.1 × 10^6^ CFU/implant at the end point of the study (120h).

Furthermore, we agree with the referee that circling the regions of interest around implanted catheters is obvious. Although we think that circles make our methods easier to understand for the reader, we replaced the circles with letters indicating where colonized and sterile catheters were implanted (C = Colonized and S = sterile), because this is how bio distribution scans are represented in literature (Karmani et al., 2016). Moreover, we added the paragraph “Image visualization and SPECT/CT data analyses” to better explain the analysis procedure.

To answer the reviewers last question, the implantation of sterile and pre-colonized catheters in the left or the right flank was randomized and the analyzing investigator was blinded for the injection of [^111^In]In-4497-IgG1 or [^111^In]In-Palivizumab. We added line 597+621 to state this in the manuscript.

• Figure 6A: the authors note that the overall antibody intensity decreases quite dramatically over 3 days. Does radiolabeling increase the turn-over of human IgG1 in mice? Or is the biofilm/infection responsible for the increased turn-over? *S. aureus* secretes several proteases that can degrade antibodies. This could be easily tested by using distinct animals for sterile and pre-colonized implants, although it is understood if adding additional animals to this particular study might be challenging.

There is ample literature available to the half-life of human/humanized IgG1 antibodies in mice. It has been described that human IgG1 usually has a half-life of 2-3 days in mice. For example, a humanized IgG1 to melanin labeled with ^111^In, displayed a 2 days half-life in C57Bl6 mice (Allen et al., 2019). Thus, significant decrease of the antibody in circulation after 72 hrs is consistent with the pharmacokinetics of human/humanized antibodies in mice. In the revised manuscript we now added this reference and better explained the dramatic decrease within the first three days. Please see line 310.

• Line 113-116: Not clear if the statement refers to the control mAbs only or also to the tested mAbs.

We replaced ‘these’ with ‘all’ to make clear that all sequences were derived from scientific or patent publications.

• Line 125: Please rephrase sentence, as it can currently be read as either "SpA complicates the detection…" or "low surface expression of SpA complicates the detection…"

We corrected this sentence to “nonspecific binding of the IgG1 Fc-domain to SpA complicates the detection”.

• Line 130-131: Please add a reference here.

There is no reference available for this strain, but construction of AH4116 is described in the methods section of the manuscript (Line 461).

• 134-135: Does this mean that (1) there is no DNA present in this biofilm, (2) the DNA is not digested by DNaseI due to shielding by other structural components (like PNAG) or (3) that digestion of DNA by DNaseI does not disturb the biofilm as other structural components (like PNAG) are present?

We thank the referee for raising this interesting question. According to literature, the PNAG vs DNA/protein phenotype are not mutually exclusive as PNAG-positive *S. aureus* biofilm has been shown to contain DNA and proteins in addition to PNAG (Mlynek et al., 2020). In Mlynek *et al.* it is suggested that PNAG can shield DNA but also that PNAG can stabilize DNA networks in a biofilm, making it less sensitive to DNase treatment. Thus, option 2 or option 3 could be possible explanations. We added these explanations to the revised manuscript in line 130.

• 148: Growth condition affects regulation of the enzymes, and thereby their presence. I would suggest rephrasing the sentence to include both (1) presence of the genes encoding the responsible glycosyltransferase enzymes, as well as (2) expression of these genes based on environmental conditions (such as growth phase). Cite e.g. Li et al. 2015 (DOI: 10.1038/srep17219) regarding point 1.

The revised manuscript now contains this reference and the sentence has been rephrased. Please see line 142-146.

• 153: This is related to the used growth phase of the bacterial cells (24h), which features high tarM expression. I would suggest also performing this experiment using exponential phase cultures, similar to how this is also performed for the ClfA experiments later on, in order to avoid any confusion regarding presence of the epitope on this strain in the literature.

We apologize to the referee that it was not described clearly that all experiments in the main figures were performed with exponential phase bacteria. Only data in Figure 4—figure supplement 1 has been obtained with stationary phase *S. aureus*. We specified more clearly throughout the revised manuscript when we are using exponential or stationary cultures.

• 161-164: See above. This is only true when comparing with stationary phase planktonic growth in broth.

In the original manuscript we compared exponential phase LAC*∆spa∆sbi* (AH4116) to biofilm. This is related to point 6 and point 17 and has been made more clear throughout the revised manuscript.

• 178-179: The authors may consider using a PNAG-pos strain with both WTA modifications and an isogenic ica knockout to substantiate this claim.

This is related to point 20 where we downgraded our claims about the location of mAb binding mAbs in the 3D structure of biofilm because this an observation and not the focus of our manuscript.

• 192-194: How was this objectively measured? By comparison of the centers of mass of the fluorescent signals? If so, it would be helpful to provide an appropriate statistical method to substantiate these claims.

We agree with the referee that these statements are rather subjective and not proven by statistics. To test the statistical significance, we now performed unpaired t-tests on the mean centers of mass of >2 Z-stacks of the red versus the green channels for n=2 independent experiments. Only in the case of 4497-IgG1 binding in Wood46 biofilm, a significant difference in the center of mass was found. Based on this analysis, we removed the Z-stack profiles (in old Figure 2ef, old Figure 3ef, Figure 2—figure supplement 3, Figure 2—figure supplement 4) from the manuscript and removed all lines describing the distribution of mAbs in the biofilm 3D structure as indicated by track changes.

• 198.201: This experiment needs some more explanation. For instance, an additional mAb A120 was used in Figure S5c, which should be described. If it is indeed WTA b-GlcNAc that is bound by CR5132, please also show deposition on exponential-phase planktonic LAC.

We thank the reviewer for pointing out that this experiment was not explained well. We added line 211-220 to explain this experiment better. Also, A120 was added to the list of antibodies (Supplementary File 1) and described in the revised manuscript text (line 213). Furthermore, we see comparable binding of CR5132-IgG3 and 4497-IgG3 to exponential planktonic wild-type LAC, data which has been included in the manuscript in Figure 5—figure supplement 4.

• 217- 219: The authors to not provide compelling evidence that the abundance of the ClfA is different between PNAG-pos and PNAG-neg biofilms. They should tone down their statement or, if they want to make such claims, should show this using a different method, e.g. western blot or on the transcriptional level by qPCR.

As suggested by the reviewer, we rephrased this line to make it more clear that we are giving two possible explanations for the differences in T1-2 binding that we observed between biofilm formed the two strains. Please see line 237.

• 233-235: What is this ratio based on? Please describe. Not taking into account kinetics, one would need to administer at least 10 g/l * 5 l / 25 = 2 gram of monoclonal antibody to reach a similar ratio in the human circulation. Is that realistic? If not, what would be a more appropriate ratio to test in vitro?

This ratio was based on the calculation given by the reviewer, where we think administration of 2 gram monoclonal antibody is realistic as there are ongoing clinical trials (SAATELLITE [NCT02296320]) where 2000 and 5000 mg intravenous mAb is being administered to participants. We added line 257 to the revised manuscript.

• 274: How do the authors conclude this, as only one other body site seems to be checked (namely, the sterile control implant)?

We completely agree with the reviewer and removed this statement from the manuscript.

• 275-276: Which data are the authors correlating to make this statement? Please test appropriately.

We thank the reviewer for raising this point. In the original manuscript, we compared the CFU counts in old figure S10B to old figure 6B, where each shape represents one mouse and it seemed that in mice with higher CFU counts on the implant at end point, we could detect a higher percentage of mAb signal at the colonized implant. In the revised manuscript, these mice can be recognized by their shape in Figure 7b, 120h and are compared to corresponding CFU counts in Figure 7—figure supplement 1b. We adjusted the text to “when a higher bacterial burden was recovered from a pre-colonized implant (n=3) at the end point (Figure 7—figure supplement 1, each shape is one mouse), a higher 4497-IgG1 activity was measured at the implant (Figure 7b, 120h, see corresponding shapes)”. Please see line 328.

• 283-285: What about S. epidermidis? Isn't that a similar if not bigger problem regarding these sorts of infection?

The referee is right that *S. epidermidis* causes similar amounts of implant infections. We focused on *S. aureus* because this often has a more serious course of disease. We know from previous publications that some mAbs (rF1 (Hazenbos et al., 2013), F598 (Kelly-Quintos et al., 2006)), CR5132 [US 2012/0141493 A1] can bind to planktonic *S. epidermidis* and hypothesize that these also bind to biofilm. Although we feel this is outside the scope of this manuscript, we included a statement about *S. epidermidis* in the discussion (line 380-383).

• References: Text duplication in all references containing "*Staphylococcus aureus*": in all cases written as "*Staphylococcus aureusStaphyloccus aureus*"

We corrected *"Staphylococcus aureusStaphyloccus aureus"* to *"Staphylococcus aureus".*

• Figures: Please plot two-sided error bars instead of the top one only. In addition, several y-axes are too short and clip error-bars (e.g. 3b), please double-check.

We corrected all figures to show two sided error bars.

• 755-756: What does control refer to? No Ab or one of the isotype control antibodies defined in Figure 1A?

In the revised manuscript we specified in the figure legend that we used 66nM of G2a-2 (anti-DNP) ctrl IgG1 here (Line 915).

References:

Allen, K. J. H., Jiao, R., Malo, M. E., Frank, C., Fisher, D. R., Rickles, D., and Dadachova, E. (2019). Comparative radioimmunotherapy of experimental melanoma with novel humanized antibody to melanin labeled with 213bismuth and 177lutetium. Pharmaceutics, 11(7). https://doi.org/10.3390/pharmaceutics11070348

Arciola, C. R., Campoccia, D., Ravaioli, S., and Montanaro, L. (2015). Polysaccharide intercellular adhesin in biofilm: Structural and regulatory aspects. Frontiers in Cellular and Infection Microbiology, 5(FEB), 1–10. https://doi.org/10.3389/fcimb.2015.00007

Branderhorst, W., Blezer, E. L. A., Houtkamp, M., Ramakers, R. M., Van Den Brakel, J. H., Witteveen, H., Van Der Have, F., Van Andel, H. A. G., Vastenhouw, B., Wu, C., Stigter-vanWalsum, M., Van Dongen, G. A. M. S., Viergever, M. A., Bleeker, W. K., and Beekman, F. J. (2014). Three-dimensional histologic validation of high-resolution spect of antibody distributions within xenografts. Journal of Nuclear Medicine. https://doi.org/10.2967/jnumed.113.125401

Fitzpatrick, F., Humphreys, H., and O’Gara, J. P. (2005). Evidence for icaADBC-independent biofilm development mechanism in methicillin-resistant *Staphylococcus aureus* clinical isolates. Journal of Clinical Microbiology, 43(4), 1973–1976. https://doi.org/10.1128/JCM.43.4.1973-1976.2005

Formosa-Dague, C., Feuillie, C., Beaussart, A., Derclaye, S., Kucharíková, S., Lasa, I., Van Dijck, P., and Dufreîne, Y. F. (2016). Sticky Matrix: Adhesion Mechanism of the Staphylococcal Polysaccharide Intercellular Adhesin. ACS Nano, 10(3), 3443–3452. https://doi.org/10.1021/acsnano.5b07515

Hazenbos, W. L. W., Kajihara, K. K., Vandlen, R., Morisaki, J. H., Lehar, S. M., Kwakkenbos, M. J., Beaumont, T., Bakker, A. Q., Phung, Q., Swem, L. R., Ramakrishnan, S., Kim, J., Xu, M., Shah, I. M., Diep, B. A., Sai, T., Sebrell, A., Khalfin, Y., Oh, A., … Mariathasan, S. (2013). Novel Staphylococcal Glycosyltransferases SdgA and SdgB Mediate Immunogenicity and Protection of Virulence-Associated Cell Wall Proteins. PLoS Pathogens, 9(10). https://doi.org/10.1371/journal.ppat.1003653

Karmani, L., Levêque, P., Bouzin, C., Bol, A., Dieu, M., Walrand, S., Vander Borght, T., Feron, O., Grégoire, V., Bonifazi, D., Michiels, C., Lucas, S., and Gallez, B. (2016). Biodistribution of 125I-labeled anti-endoglin antibody using SPECT/CT imaging: Impact of in vivo deiodination on tumor accumulation in mice. Nuclear Medicine and Biology. https://doi.org/10.1016/j.nucmedbio.2016.03.007

Kelly-Quintos, C., Cavacini, L. A., Posner, M. R., Goldmann, D., and Pier, G. B. (2006). Characterization of the opsonic and protective activity against *Staphylococcus aureus* of fully human monoclonal antibodies specific for the bacterial surface polysaccharide poly-N-acetylglucosamine. Infection and Immunity, 74(5), 2742–2750. https://doi.org/10.1128/IAI.74.5.2742-2750.2006

McCarthy, H., Rudkin, J. K., Black, N. S., Gallagher, L., O’Neill, E., and O’Gara, J. P. (2015). Methicillin resistance and the biofilm phenotype in *Staphylococcus aureus*. Frontiers in Cellular and Infection Microbiology, 5(January), 1–9. https://doi.org/10.3389/fcimb.2015.00001

Mlynek, K. D., Bulock, L. L., Stone, C. J., Curran, L. J., Sadykov, M. R., Bayles, K. W., and Brinsmade, S. R. (2020). Genetic and biochemical analysis of cody-mediated cell aggregation in *Staphylococcus aureus* reveals an interaction between extracellular DNA and polysaccharide in the extracellular matrix. Journal of Bacteriology, 202(8), 1–21. https://doi.org/10.1128/JB.00593-19

O’Neill, E., Pozzi, C., Houston, P., Smyth, D., Humphreys, H., Robinson, D. A., and O’Gara, J. P. (2007). Association between methicillin susceptibility and biofilm regulation in *Staphylococcus aureus* isolates from device-related infections. Journal of Clinical Microbiology, 45(5), 1379–1388. https://doi.org/10.1128/JCM.02280-06

Rohde, H., Burandt, E. C., Siemssen, N., Frommelt, L., Burdelski, C., Wurster, S., Scherpe, S., Davies, A. P., Harris, L. G., Horstkotte, M. A., Knobloch, J. K. M., Ragunath, C., Kaplan, J. B., and Mack, D. (2007). Polysaccharide intercellular adhesin or protein factors in biofilm accumulation of Staphylococcus epidermidis and *Staphylococcus aureus* isolated from prosthetic hip and knee joint infections. Biomaterials, 28(9), 1711–1720. https://doi.org/10.1016/j.biomaterials.2006.11.046

Sugimoto, S., Sato, F., Miyakawa, R., Chiba, A., Onodera, S., Hori, S., and Mizunoe, Y. (2018). Broad impact of extracellular DNA on biofilm formation by clinically isolated Methicillin-resistant and -sensitive strains of *Staphylococcus aureus*. Scientific Reports, 8(1). https://doi.org/10.1038/s41598-018-20485-z

Zapotoczna, M., McCarthy, H., Rudkin, J. K., O’Gara, J. P., and O’Neill, E. (2015). An essential role for coagulase in *Staphylococcus aureus* biofilm development reveals new therapeutic possibilities for device-related infections. Journal of Infectious Diseases, 212(12), 1883–1893. https://doi.org/10.1093/infdis/jiv319